# Identifiable Contrastive Learning with Automatic Feature Importance Discovery

**Qi Zhang**[1]\* **Yifei Wang**[2]\* **Yisen Wang**[1,3]†

[1] National Key Lab of General Artificial Intelligence,
School of Intelligence Science and Technology, Peking University
[2] School of Mathematical Sciences, Peking University
[3] Institute for Artificial Intelligence, Peking University

## Abstract

Existing contrastive learning methods rely on pairwise sample contrast $z_x^\top z_{x'}$ to learn data representations, but the learned features often lack clear interpretability from a human perspective. Theoretically, it lacks feature identifiability and different initialization may lead to totally different features. In this paper, we study a new method named tri-factor contrastive learning (triCL) that involves a 3-factor contrast in the form of $z_x^\top S z_{x'}$, where $S = \operatorname{diag}(s_1, \ldots, s_k)$ is a learnable diagonal matrix that automatically captures the importance of each feature. We show that by this simple extension, triCL can not only obtain identifiable features that eliminate randomness but also obtain more interpretable features that are ordered according to the importance matrix $S$. We show that features with high importance have nice interpretability by capturing common classwise features, and obtain superior performance when evaluated for image retrieval using a few features. The proposed triCL objective is general and can be applied to different contrastive learning methods like SimCLR and CLIP. We believe that it is a better alternative to existing 2-factor contrastive learning by improving its identifiability and interpretability with minimal overhead. Code is available at https://github.com/PKU-ML/Tri-factor-Contrastive-Learning.

## 1 Introduction

As a representative self-supervised paradigm, contrastive learning obtains meaningful representations and achieves state-of-the-art performance in various tasks by maximizing the feature similarity $z_x^\top z_x^+$ between samples augmented from the same images while minimizing the similarity $z_x^\top z_x^-$ between independent samples [4, 20, 15, 5, 18]. Besides the empirical success, recent works also discuss the theoretical properties and the generalization performance of contrastive learning [32, 30, 18].

However, there still exist many properties of contrastive learning that are not guaranteed. In this paper, we focus on a significant one: the feature identifiability. Feature identifiability in the representation learning refers to the property there exists a single, global optimal solution to the learning objective. Consequently, the learned representations can be reproducible regardless of the initialization and the optimizing procedure is useful. As a well-studied topic, identifiability is a desirable property for various tasks, including but not limited to, transfer learning [10], fair classification [26] and causal inference [24]. The previous works propose that contrastive learning obtains the linear feature identifiability while lacking exact feature identifiability, i.e., the optimal solutions obtain a freedom of linear transformations [29]. As a result, the different features are coupled, which hurts the interpretability and performance of learned representations.

---

\*Equal Contribution.
†Corresponding Author: Yisen Wang (yisen.wang@pku.edu.cn).

37th Conference on Neural Information Processing Systems (NeurIPS 2023).

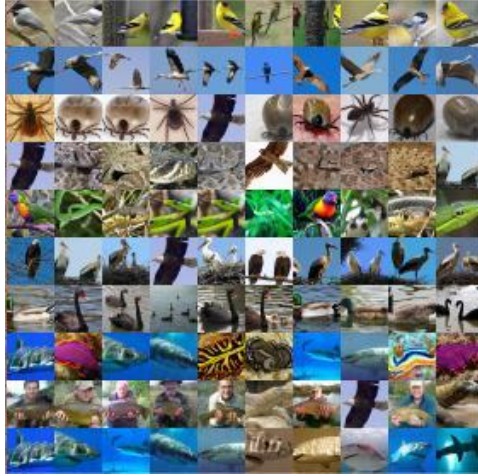 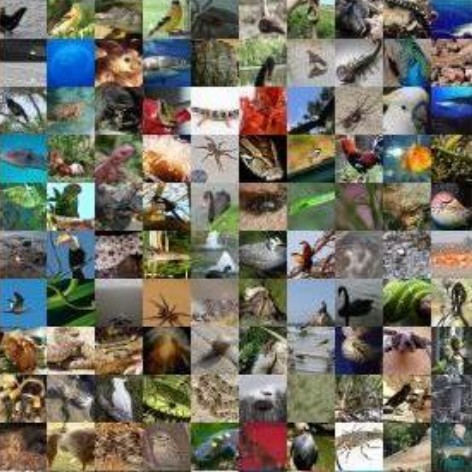



(a) 10 most important dimensions        (b) 10 least important dimensions



Figure 1: The visualization of samples on ImageNet-100 that have the largest values in 10 selected dimensions of representations learned by tri-factor contrastive learning (each row represents a dimension).

In this paper, we propose a new contrastive learning model: tri-factor contrastive learning (triCL), which introduces a 3-factor contrastive loss, i.e., we replace $z_x^\top z_{x'}$ with $z_x^\top S z_{x'}$ when calculating the similarity between two features, where $S$ is a learnable diagonal matrix called importance matrix. We theoretically prove that triCL absorbs the freedom of linear transformations and enables **exact feature identifiability**. Besides, we observe that triCL shows other satisfying properties. For example, the generalization performance of triCL is theoretically guaranteed. What is more, we find that the diagonal values of the importance matrix $S$ in triCL indicate the degrees of feature importance. In Figure 1, we visualize the samples that have the largest values in the most and least important dimensions ordered by the importance matrix. We find the samples activated in the more important dimensions are more semantically similar, which verifies the order of feature importance in triCL is quite close to the ground truth. Theoretically, we prove that the dimensions related to the larger values in the importance matrix make more contributions to decreasing the triCL loss. With the automatic discovery of feature importance in triCL, the downstream task conducted on the representations can be accelerated by selecting the important features and throwing the meaningless features.

As triCL is a quite simple and general extension, we apply it to different contrastive learning frameworks, such as SimCLR [4] and CLIP [28]. Empirically, we first verify the identifiability of triCL and further evaluate the performance of triCL on real-world datasets including CIFAR-10, CIFAR-100, and ImageNet-100. In particular, with the automatic discovery of important features, triCL demonstrates significantly better performance on the downstream tasks using a few feature dimensions. We summarize our contributions as follows:

- We propose tri-factor contrastive learning (triCL), the first contrastive learning algorithm which enables exact feature identifiability. Additionally, we extend triCL to different contrastive learning methods, such as spectral contrastive learning, SimCLR and CLIP.

- Besides the feature identifiability, we analyze several theoretical properties of triCL. Specifically, we construct the generalization guarantee for triCL and provide theoretical evidence that triCL can automatically discover the feature importance.

- Empirically, we verify that triCL enables the exact feature identifiability and triCL can discover the feature importance on the synthetic and real-world datasets. Moreover, we investigate whether triCL obtains superior performance in downstream tasks with selected representations.

## 2 Related Work

**Self-supervised Learning.** Recently, to get rid of the expensive cost of the labeled data, self-supervised learning has risen to be a promising paradigm for learning meaningful representations by designing various pretraining tasks, including context-based tasks [14], contrastive learning [20] and masked image modeling [21]. Among them, contrastive learning is a popular algorithm that achieves impressive success and largely closes the gap between self-supervised learning and supervised learning [27, 4, 20, 33]. The idea of contrastive learning is quite simple, i.e., pulling the semantically similar samples (positive samples) together while pushing the dissimilar samples (negative samples) away in the feature space. To better achieve this objective, recent works propose different variants of contrastive learning, such as introducing different training objectives [36, 18, 35], different structures [4, 6, 16, 39], different sampling processes [11, 25, 38, 7] and different empirical tricks [20, 3].

**Theoretical Understandings of Contrastive Learning.** Despite the empirical success of contrastive learning, the theoretical understanding of it is still limited. [30] establish the first theoretical guarantee on the downstream classification performance of contrastive learning by connecting the InfoNCE loss and Cross Entropy loss. As there exists some unpractical assumptions in the theoretical analysis in [30], recent works improve the theoretical framework and propose new bounds [1, 2, 34]. Moreover, [18] analyzes the downstream performance of contrastive learning from a graph perspective and constructs the connection between contrastive learning and spectral decomposition. Recently, researchers have taken the inductive bias in contrastive learning into the theoretical framework and shown the influence of different network architectures [17, 31]. Except for the downstream classification performance of contrastive learning, some works focus on other properties of representations learned by contrastive learning. [23] discuss the feature diversity by analyzing the dimensional collapse in contrastive learning. [29] prove that the contrastive models are identifiable up to linear transformations under certain assumptions.

## 3 Preliminary

**Contrastive Pretraining Process.** We begin by introducing the basic notations of contrastive learning. The set of all natural data is denoted as $\mathcal{D}_u = \{\bar{x}_i\}_{i=1}^{N_u}$ with distribution $\mathcal{P}_u$, and each natural data $\bar{x} \in \mathcal{D}_u$ has a ground-truth label $y(\bar{x})$. An augmented sample $x$ is generated by transforming a natural sample $\bar{x}$ with the augmentations distributed with $\mathcal{A}(\cdot|\bar{x})$. The set of all the augmented samples is denoted as $\mathcal{D} = \{x_i\}_{i=1}^{N}$. We assume both sets of natural samples and augmented samples to be finite but exponentially large to avoid non-essential nuances in the theoretical analysis and it can be easily extended to the case where they are infinite [18]. During the pretraining process, we first draw a natural sample $\bar{x} \sim \mathcal{P}_u$, and independently generate two augmented samples $x \sim \mathcal{A}(\cdot|\bar{x})$, $x^+ \sim \mathcal{A}(\cdot|\bar{x})$ to construct a positive pair $(x, x^+)$. For the negative samples, we independently draw another natural sample $\bar{x}^- \sim \mathcal{P}_u$ and generate $x^- \sim \mathcal{A}(\cdot|\bar{x}^-)$. With positive and negative pairs, we learn the encoder $f : \mathbb{R}^d \to \mathbb{R}^k$ with the contrastive loss. For the ease of our analysis, we take the spectral contrastive loss [18] as an example:

$$\mathcal{L}_{\mathrm{SCL}}(f) = -2\mathbb{E}_{x,x^+} f(x)^\top f(x^+) + \mathbb{E}_x \mathbb{E}_{x^-} (f(x)^\top f(x^-))^2. \tag{1}$$

We denote $z_x = f(x)$ as the features encoded by the encoder. By optimizing the spectral loss, the features of positive pairs $(z_x^\top z_{x^+})$ are pulled together while the negative pairs $(z_x^\top z_{x^-})$ are pushed apart.

**Augmentation Graph.** A useful theoretical framework to describe the properties of contrastive learning is to model the learning process from the augmentation graph perspective [18]. The augmentation graph is defined over the set of augmented samples $\mathcal{D}$, with its adjacent matrix denoted by $A$. In the augmentation graph, each node corresponds to an augmented sample, and the weight of the edge connecting two nodes $x$ and $x^+$ is equal to the probability that they are selected as a positive pair, i.e., $A_{xx^+} = \mathbb{E}_{\bar{x} \sim \mathcal{P}_u} [\mathcal{A}(x|\bar{x})\mathcal{A}(x^+|\bar{x})]$. And we denote $\bar{A}$ as the normalized adjacent matrix of the augmentation graph, i.e., $\bar{A} = D^{-1/2}AD^{-1/2}$, where $D$ is a diagonal matrix and $D_{xx} = \sum_{x' \in \mathcal{D}} A_{xx'}$. To analyze the properties of $\bar{A}$, we denote $\bar{A} = U\Sigma V^\top$ is the singular value decomposition (SVD) of the normalized adjacent matrix $\bar{A}$, where $U \in \mathbb{R}^{N \times N}, V \in \mathbb{R}^{N \times N}$ are unitary matrices, and $\Sigma = \mathrm{diag}(\sigma_1, \ldots, \sigma_N)$ contains descending singular values $\sigma_1 \geq \ldots \sigma_N \geq 0$.

# 4 Exact Feature Identifiability with Tri-factor Contrastive Learning

In this section, we propose a new representation learning paradigm called tri-factor contrastive learning (triCL) that enables exact feature identifiability in contrastive learning. In Section 4.1, we prove that contrastive learning obtains linear identifiability, i.e., the freedom in the optimal solutions are linear transformations. In Section 4.2, we introduce the learning process of triCL and theoretically verify it enables the exact feature identifiability.

## 4.1 Feature Identifiability of Contrastive Learning

When using a pretrained encoder for downstream tasks, it is useful if the learned features are reproducible, in the sense that when the neural network learns the representation function on the same data distribution multiple times, the resulting features should be approximately the same. For example, reproducibility can enhance the interpretability and the robustness of learned representations [22]. One rigorous way to ensure reproducibility is to select a model whose representation function is *identifiable in function space* [29]. To explore the *feature identifiability* of contrastive learning, we first characterize its general solution.

**Lemma 4.1** ([18]). *Let $\bar{A} = U\Sigma V^\top$ be the SVD decomposition of the normalized adjacent matrix $\bar{A}$. Assume the neural networks are expressive enough for any features. The spectral contrastive loss (Eq. 1) attains its optimum when $\forall\, x \in \mathcal{D}$,*

$$f^*(x) = \frac{1}{\sqrt{D_{xx}}} \left( U_x^k \operatorname{diag}(\sigma_1, \ldots, \sigma_k) R \right)^\top, \tag{2}$$

*where $U_x$ takes the $x$-th row of $U$, $U^k$ denotes the submatrices containing the first $k$ columns of $U$, and $R \in \mathbb{R}^{k \times k}$ is an arbitrary unitary matrix.*

From Lemma 4.1, we know that the optimal representations are not unique, due to the freedom of affine transformations. This is also regarded as a relaxed notion of feature identifiability, named *linear feature identifiability* $\overset{L}{\sim}$ defined below [29].

**Definition 4.2** (Linear feature identifiability). Let $\overset{L}{\sim}$ be a pairwise relation in the encoder function space $\mathcal{F} = \{f : \mathcal{X} \to \mathbb{R}^k\}$ defined as:

$$f' \overset{L}{\sim} f^* \iff f'(x) = Af^\star(x), \forall\, x \in \mathcal{X}, \tag{3}$$

where $A$ is an invertible $k \times k$ matrix.

It is apparent that the optimal encoder $f$ (Eq. 2) obtained from the spectral contrastive loss (Eq. 1) is linearly identifiable. Nevertheless, there are still some ambiguities *w.r.t.* linear transformations in the model. Although the freedom of linear transformations can be absorbed on the linear probing task [18], these representations may show varied results in many downstream tasks e.g., in the k-NN evaluation process. So we wonder whether we could achieve the exact feature identifiability. We first further define two kinds of more accurate feature identifiabilities below.

**Definition 4.3** (Sign feature identifiability). Let $\overset{S}{\sim}$ be a pairwise relation in the encoder function space $\mathcal{F} = \{f : \mathcal{X} \to \mathbb{R}^k\}$ defined as:

$$f' \overset{S}{\sim} f^* \iff f'_j(x) = \pm f_j^\star(x), \forall\, x \in \mathcal{X}, j \in [k]. \tag{4}$$

where $f_j(x)$ is the $j$-th dimension of $f(x)$.

**Definition 4.4** (exact feature identifiability). Let $\sim$ be a pairwise relation in the encoder function space $\mathcal{F} = \{f : \mathcal{X} \to \mathbb{R}^k\}$ defined as:

$$f' \sim f^* \iff f'(x) = f^\star(x), \forall\, x \in \mathcal{X}. \tag{5}$$

## 4.2 Tri-factor Contrastive Learning with Exact Feature Identifiability

Motivated by the trifactorization technique in matrix decomposition problems [9] and the equivalence between the spectral contrastive loss and the matrix decomposition objective [18], we consider adding a learnable diagonal matrix when calculating the feature similarity in the contrastive loss to absorb

the freedom of linear transformations. To be specific, we introduce a contrastive learning model that enables exact feature identifiability, named *tri-factor contrastive learning (triCL)*, which adopts a tri-term contrastive loss:

$$\mathcal{L}_{\text{tri}}(f, S) = -2\mathbb{E}_{x,x^+} f(x)^\top S f(x^+) + \mathbb{E}_x \mathbb{E}_{x^-} \left( f(x)^\top S f(x^-) \right)^2, \tag{6}$$

where we name $S = \text{diag}(s_1, \ldots, s_k)$ as the importance matrix and it is a diagonal matrix with $k$ non-negative *learnable* parameters satisfying $s_1 \geq \cdots \geq s_k \geq 0$ [3]. Additionally, the encoder $f$ is constrained to be decorrelated, *i.e.,*

$$\mathbb{E}_x f_i(x)^\top f_j(x) = \begin{cases} 1, & \text{if } i = j \\ 0, & \text{if } i \neq j \end{cases}, \; i, j \in [k], \tag{7}$$

for an encoder $f : \mathbb{R}^d \to \mathbb{R}^k$. One way to ensure feature decorrelation is the following penalty loss,

$$\mathcal{L}_{\text{dec}}(f) = \left\| \mathbb{E}_x f(x)f(x)^\top - I \right\|^2, \tag{8}$$

leading to a combined triCL objective,

$$\mathcal{L}_{\text{triCL}}(f, S) = \mathcal{L}_{\text{tri}}(f, S) + \mathcal{L}_{\text{dec}}(f). \tag{9}$$

Similar feature decorrelation objectives have been proposed in non-contrastive visual learning methods with slightly different forms, *e.g.,* Barlow Twins [36].

Since triCL automatically learns feature importance $S$ during training, it admits a straightforward feature selection approach. Specifically, if we need to select $m$ out of $k$ feature dimensions for downstream tasks (*e.g.,* in-time image retrieval), we can sort the feature dimensions according to their importance $s_i$'s (after training), and simply use the top $m$ features as the most important ones. Without loss of generality, we assume $s_1 \geq \cdots \geq s_k$, and the top $m$ features are denoted as $f^{(m)}$.

**Identifiability of TriCL.** In the following theorem, we show that by incorporating the diagonal importance matrix $S$ that regularizes features along each dimension, triCL can resolve the linear ambiguity of contrastive learning and become sign-identifiable.

**Theorem 4.5.** *Assume the normalized adjacent matrix $\bar{A}$ has distinct largest $k$ singular values ($\forall\, i, j \in [k]$, $\sigma_i \neq \sigma_j$ when $i \neq j$) and the neural networks are expressive enough, the tri-factor contrastive learning (triCL, Eq. 9) attains its optimum when $\forall\, x \in \mathcal{D}, j \in [k]$*

$$f_j^\star(x) = \pm \frac{1}{\sqrt{D_{xx}}} \left( U_x^k \right)_j, S^* = \text{diag}(\sigma_1, \ldots, \sigma_k), \tag{10}$$

*which states that the tri-factor contrastive learning enables the **sign feature identifiability**.*

As shown in Theorem 4.5, the only difference remaining in the solutions (Eq. 10) is the sign. To remove this ambiguity, for dimension $j$, we randomly select a natural sample $\bar{x} \sim \mathcal{P}_u$, encode it with the optimal solution $f^\star$ of triCL, and observe the sign of its $j$-th dimension $f_j^\star(\bar{x})$. If $f_j^\star(\bar{x}) = 0$, we draw another sample and repeat the process until we obtain a non-zero feature $f_j^\star(\bar{x})$. We then store the sample as an original point $x_{0j}$ and adjust the sign of different learned representations as follows:

$$\bar{f}_j^\star(x) = (-1)^{\mathbb{1}(f_j^\star(x_{0j})>0)} \cdot f_j^\star(x), \forall x \in \mathcal{D}, j \in [k] \tag{11}$$

By removing the freedom of sign, the solution becomes unique and triCL enables the exact feature identifiability:

**Corollary 4.6.** *Set $\bar{f}^\star$ as the final learned encoder of tri-factor contrastive learning, and then triCL obtains the exact feature identifiability.*

## 5 Theoretical Properties of Tri-factor Contrastive Learning

Besides the feature identifiability, we analyze other theoretical properties of triCL in this section. Specifically, in Section 5.1, we provide the generalization guarantee of triCL and we present another advantage of triCL: triCL can automatically discover the importance of different features. In Section 5.2, we extend triCL to other contrastive learning frameworks.

---

[3]In practice, we enforce the non-negative conditions by applying the softplus activation functions on the diagonal values of $S$. We only enforce the monotonicity at the *end* of training by simply sorting different rows of $S$ and different dimensions of $f(x)$ by the descending order of corresponding diagonal values in $S$.

## 5.1 Downstream Generalization of Tri-factor Contrastive Learning

In the last section, we propose a new contrastive model (triCL) that enables exact feature identifiability. In the next step, we aim to theoretically discuss the downstream performance of the identifiable representations learned by triCL. For the ease of our theoretical analysis, we first focus on a common downstream task: Linear Probing. Specifically, we denote the linear probing error as the classification error of the optimal linear classifier on the pretrained representations, i.e., $\mathcal{E}(f) = \min_g \mathbb{E}_{\bar{x} \sim \mathcal{P}_u} \mathbb{1}[g(f(\bar{x})) \neq y(\bar{x})]$. By analyzing the optimal solutions of triCL when using the full or partial features, we obtain the following theorem characterizing their downstream performance.

**Theorem 5.1.** *We denote $\alpha$ as the probability that the natural samples and augmented views have different labels, i.e., $\alpha = \mathbb{E}_{\bar{x} \sim \mathcal{P}_u} \mathbb{E}_{x \sim \mathcal{A}(\cdot|\bar{x})} \mathbb{1}[y(\bar{x}) \neq y(x)]$. Let $f_{SCL}^{\star}$ and $(f_{triCL}^{\star}, S^{\star})$ be the optimal solutions of SCL and triCL, respectively. Then, SCL and triCL have the same downstream classification error when using all features*

$$\mathcal{E}(f_{SCL}^{\star}) = \mathcal{E}((f_{triCL}^{\star})) \leq c_1 \sum_{i=k+1}^{N} \sigma_i^2 + c_2 \cdot \alpha, \tag{12}$$

*where $\sigma_i$ is the $i$-th largest eigenvalue of $\bar{A}$, and $c_1, c_2$ are constants. When using only $m \leq k$ features of the optimal features, triCL with the top $m$ features admits the following error bound*

$$\mathcal{E}(f_{triCL}^{\star(m)}) \leq c_1 \sum_{i=m+1}^{N} \sigma_i^2 + c_2 \cdot \alpha := U(f_{triCL}^{\star(m)}). \tag{13}$$

*Instead, without importance information, we can only randomly select $m$ SCL features (denoted as $f_{SCL}^{(m)}$), which has the following error bound (taking expectation over all random choices)*

$$\mathcal{E}(f_{SCL}^{\star(m)}) \leq c_1 \left( (1 - \frac{m}{k}) \sum_{i=1}^{k} \sigma_i^2 + \sum_{i=k+1}^{N} \sigma_i^2 \right) + c_2 \cdot \alpha := U(f_{SCL}^{\star(m)}). \tag{14}$$

*Comparing the two upper bounds, we can easily conclude that*

$$U(f_{SCL}^{\star(m)}) - U(f_{triCL}^{\star(m)}) \geq \frac{m(k-m)}{k} (\frac{1}{m} \sum_{i=1}^{m} \sigma_i^2 - \frac{1}{k-m} \sum_{i=m+1}^{k} \sigma_i^2) \geq 0. \tag{15}$$

*Thus, triCL admits a smaller error when using a subset of features for downstream classification.*

Theorem 5.1 shows that triCL is particularly helpful for downstream tasks when we select a subset of features according to the learned feature importance. When using all features, the two methods converge to the same downstream error bound, which also aligns with our observations in practice.

## 5.2 Extensions to Other Contrastive Learning Frameworks

In the above sections, we propose tri-factor contrastive learning (triCL) which enables the exact feature identifiability and obtains an ordered representation while achieving the guaranteed downstream performance. In the next step, we extend the triCL to a unified contrastive learning paradigm that can be applied in different contrastive learning frameworks.

**Extension to Other SSL Methods.** As replacing the 2-factor contrast $f(x)^\top f(x')$ with the 3-factor contrast $f(x)^\top S f(x')$ is a simple operation and not constrained to the special form of spectral contrastive loss, we extend triCL to other contrastive frameworks. We first take another representative contrastive learning method SimCLR [4] as an example. Comparing the InfoNCE loss in SimCLR and the spectral contrastive loss, we find they are quite similar and the only difference is that they push away the negative pairs with different loss functions ($l_2$ loss v.s. $logsumexp$ loss). So we propose the tri-InfoNCE loss by changing the terms of negative samples in triCL (Eq. 9) to a tri-logsumexp term, i.e.,

$$\mathcal{L}_{\text{triNCE}}(f) = -\mathbb{E}_{x,x^+} \log \frac{\exp\left(f(x)^\top S f(x^+)\right)}{\mathbb{E}_{x^-} \exp(f(x)^\top S f(x^-))} + \left\| \mathbb{E}_x f(x) f(x)^\top - I \right\|^2. \tag{16}$$

Besides the InfoNCE loss used in SimCLR, we present more types of tri-factor contrastive loss in Appendix C. In summary, triCL can be applied in most of the contrastive frameworks by replacing the $f(x)^\top f(x')$ term with a tri-term $f(x)^\top S f(x')$ when calculating the similarity, where the importance matrix $S$ is a diagonal matrix capturing the feature importance.

**Extension to the Multi-modal Domain.** Different from the symmetric contrastive learning frameworks like SCL [18] and SimCLR [4], the multi-modal contrastive frameworks like CLIP [28] have asymmetric networks to encode asymmetric image-text pairs. So we extend the tri-factor contrastive loss to a unified asymmetric form:

$$
\begin{aligned}
\mathcal{L}_{\text{triCLIP}}(f_A, f_B, S) = &- 2\mathbb{E}_{x_a,x_b} f_A(x_a)^\top S f_B(x_b) + \mathbb{E}_{x_a^-,x_b^-}\left(f_A(x_a^-)^\top S f_B(x_b^-)\right)^2 \\
&+ \left\|\mathbb{E}_{x_a} f_A(x_a) f_A(x_a)^\top - I\right\|^2 + \left\|\mathbb{E}_{x_b} f_B(x_b) f_B(x_b)^\top - I\right\|^2 .
\end{aligned}
\tag{17}
$$

where $f_A, f_B$ are two different encoders with the same output dimension $k$. Uniformly, we denote that the positive pairs $(x_a, x_b) \sim \mathcal{P}_O$ and the negative samples $x_a^-, x_b^-$ are independently drawn from $\mathcal{P}_A, \mathcal{P}_B$. With different concrete definitions of $\mathcal{P}_O, \mathcal{P}_A, \mathcal{P}_B$, different representation learning paradigms can be analyzed together. For single-modal contrastive learning, the symmetric process in Section 3 is a special case of the asymmetric objective. For multi-modal learning, we denote $\mathcal{P}_A, \mathcal{P}_B$ as the distributions of different domain data and $\mathcal{P}_O$ as the joint distribution of semantically related pairs (e.g., semantically similar image-text pairs in CLIP). We theoretically prove that the asymmetric tri-factor contrastive learning still obtains the exact feature identifiability and guaranteed generalization performance in Appendix A.

# 6    Experiments

In this section, we provide empirical evidence to support the effectiveness of tri-factor contrastive learning. In section 6.1, we empirically verify that the tri-factor contrastive learning can enable the exact feature identifiability on the synthetic dataset. In section 6.2, we empirically analyze the properties of the importance matrix and demonstrate the advantages of automatic feature importance discovery on various tasks. Furthermore, in section 6.3, we empirically compare the generalization performance of tri-factor contrastive learning with different baselines including SimCLR [4] and spectral contrastive learning (SCL) [18] on CIFAR-10, CIFAR-100 and ImageNet-100.

## 6.1    The Verification of the Identifiability on the Synthetic Dataset

Due to the randomness of optimization algorithms like SGD, we usually can not obtain the optimal encoder of the contrastive loss. So we consider verifying the feature identifiability on the synthetic dataset. To be specific, we first construct a random matrix $\bar{A}$ with size $5000 \times 3000$ to simulate the augmentation graph. Then we compare two different objectives: $\|\bar{A} - FG^\top\|_F^2, \|\bar{A} - FSG^\top\|_F^2$, where $F$ is a matrix with size $5000 \times 256$, $G$ is a matrix with size $3000 \times 256$ and $S$ is a diagonal with size $256 \times 256$. With the analysis on [18], these two objectives are respectively equivalent to the spectral contrastive loss and tri-factor contrastive loss. According to the Eckart-Young Theorem [12], we obtain the optimal solutions of them with the SVD algorithms. For two objectives, we respectively obtain 10 optimal solutions and we calculate the average Euclidean distance between 10 solutions. We investigate that the optimal solutions of the trifactorization objective are equal (average distance is 0) while there exist significant differences between optimal solutions of bifactorization (average distance is 101.7891 and the variance of distance is 17.2798), which empirically verifies the optimal solutions of contrastive learning obtain the freedom while triCL can remove it. More details can be found in Appendix B.

## 6.2    The Automatic Discovery of Feature Importance

Except for the visualization example in Figure 1, we further quantitatively explore the properties of ordered representations learned by triCL.

**The Distribution of the Importance Matrix.** We first observe the distribution of the diagonal values in the learned importance matrix $S$. Specifically, we pretrain the ResNet-18 on CIFAR-10, CIFAR-100 and ImageNet-100 [8] by triCL. Then we normalize the importance matrices and ensure the sum of the diagonal values is 1. In Figure 2(a), we present the diagonal values of the importance

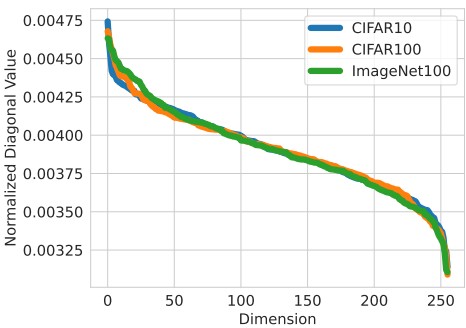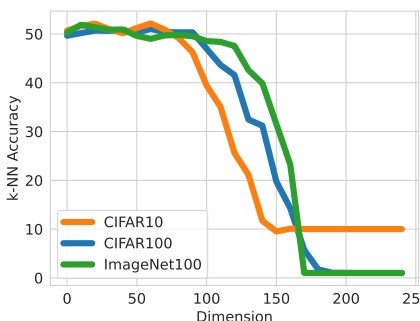

(a) The distributions of diagonal values on the importance matrix learned on different datasets.

(b) K-NN accuracy on adjacent 10 dimensions ordered by the importance matrix learned on CIFAR-10, CIFAR-100 and ImageNet-100.

Figure 2: The distributions of the discovered feature importance and the verifications on whether the order of feature importance indicated by the importance matrix is accurate.

matrices trained on different datasets and we find they are highly related to the properties of the datasets. For instance, CIFAR-100 and ImageNet-100, which possess a greater diversity of features, exhibit more active diagonal values in the top 50 dimensions compared to CIFAR-10. While for the smallest diagonal values, the different importance matrices are quite similar as most of them represent meaningless noise.

**The K-NN Accuracy on Selected Dimensions.** With the ResNet-18 and the projector pretrained on CIFAR-10, CIFAR-100 and ImageNet-100, we first sort the dimensions according to the importance matrix $S$. Then we conduct the k-NN evaluation for every 10 dimensions. As shown in Figure 2(b), the k-NN accuracy decreases slowly at first as these dimensions are related to the ground-truth labels and they share a similar significance. Then the k-NN accuracy drops sharply and the least important dimensions almost make no contributions to clustering the semantically similar samples. Note that the k-NN accuracy drops more quickly on CIFAR-10, which is consistent with the fact it has fewer types of objects.

**Linear Evaluation on Selected Dimensions.** We first train the ResNet-18 with triCL and spectral contrastive learning (SCL) on CIFAR-10. Then we select 20 dimensions from the learned representations (containing both the backbone and the projector). For triCL, we sort the dimensions according to the descending order of the importance matrix and select the largest 20 dimensions (1-20 dimensions), middle 20 dimensions (119-138 dimensions), and smallest 20 dimensions (237-256 dimensions). And for SCL, we randomly choose 20 dimensions. Then we train a linear classifier following the frozen representations with the default settings of linear probing. As shown in Figure 3(a), the linear accuracy decreases in the descending order of dimensions in triCL and the linear accuracy of the largest 20 dimensions of triCL is significantly higher than random 20 dimensions of SCL, which verifies that triCL can discover the most important semantic features related to the ground-truth labels.

**Image Retrieval.** We conduct the image retrieval on ImageNet-100. For each sample, we first encode it with the pretrained networks and then select dimensions from the features. For the methods learned by triCL, we sort the dimensions according to the values in the importance matrix $S$ and select the largest dimensions. For SCL, we randomly choose the dimensions. Then we find 100 images that have the largest cosine similarity with the query image and calculate the mean average precision (mAP) that returned images belong to the same class as the query ones. As shown in Figure 3(b), we observe the 50 dimensions of triCL show the comparable performance as the complete representation while the performance of SCL continues increasing with more dimensions, which further verifies that triCL can find the most important features that are useful in downstream tasks.

**Out-of-Distribution Generalization.** Besides the in-domain downstream tasks, we also examine whether the importance matrix can sort the feature importance accurately with out-of-domain shifts. To be specific, we prertain the ResNet-18 with SCL and triCL on ImageNet-100 and then conduct

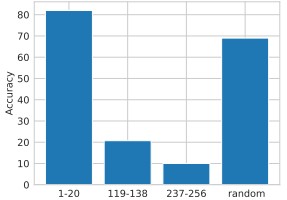

(a) Linear evaluation results of selected dimensions of triCL and SCL on CIFAR-10.

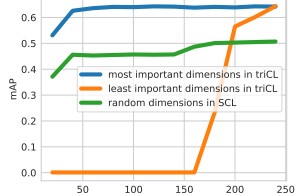

(b) The mean average precision in the image retrieval on the selected dimension of triCL and SCL on ImageNet-100.

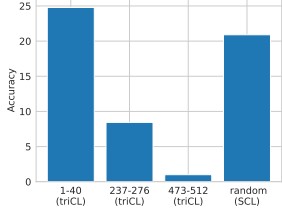

(c) Transfer accuracy of selected dimensions of triCL and SCL on stylized ImageNet-100.

Figure 3: With the automatic discovery of feature importance, Tri-factor contrastive learning (triCL) obtains superior performance than spectral contrastive learning (SCL) on downstream tasks that use selected representations.

Table 1: The linear probing acuuracy and finetune accuracy of ResNet-18 pretrained by tri-factor contrastive learning and self-supervised baselines on CIFAR-10, CIFAR-100, ImageNet-100. TriCL averagely obtains a comparable performance as the original contrastive learning. Moreover, triCL achieves significant improvements on some tasks, such as finetuning on CIFAR-100.

|  | Linear Accuracy | | | Finetune Accuracy | | |
|---|---|---|---|---|---|---|
|  | CIFAR10 | CIFAR100 | ImageNet100 | CIFAR10 | CIFAR100 | ImageNet100 |
| SCL | **88.4** | 60.3 | **72.3** | 92.3 | 66.1 | **76.4** |
| tri-SCL | 88.3 | **60.8** | 71.2 | **92.7** | **67.8** | 76.3 |
| SimCLR | **87.9** | **60.2** | 73.5 | 92.2 | 72.5 | **75.9** |
| tri-SimCLR | **87.9** | 59.8 | **74.1** | **92.3** | **73.3** | 75.6 |

linear probing with 40 dimensions of learned representations on the out-of-domain dataset stylized ImageNet-100 [13]. During the downstream evaluation process, we sort the dimensions according to the descending order of the importance matrix and select the largest 40 dimensions (1-40 dimensions), middle 40 dimensions (237-276 dimensions), and smallest 40 dimensions (473-512 dimensions) of the features learned by triCL. And for SCL, we randomly select 40 dimensions. As shown in Figure 3(c), we observe that the transfer accuracy of the most important features learned by triCL shows significantly superior performance, which shows the robustness and effectiveness of the importance matrix.

## 6.3 Transfer Learning on Benchmark Datasets

**Setups.** During the pretraining process, we utilize ResNet-18 [19] as the backbone and train the models on CIFAR-10, CIFAR-100 and ImageNet-100 [8]. We pretrain the model for 200 epochs On CIFAR-10, CIFAR-100, and for 400 epochs on ImageNet-100. We select two self-supervised methods as our baselines and apply tri-factor contrastive learning on them, including spectral contrastive learning (SCL) [18], SimCLR [4]. When implementing the tri-factor contrastive learning, we follow the default settings of the baseline methods. During the evaluation process, we consider two transfer learning tasks: linear evaluation and finetuning. During the linear evaluation, we train a classifier following the frozen backbone pretrained by different methods for 50 epochs. During the finetuning process, we train the whole network (including the backbone and the classifier) for 30 epochs.

**Results.** As shown in Table 1, we find that triCL shows comparable performance as the original contrastive learning methods in transfer learning on different real-world datasets, which is consistent with our theoretical analysis. Meanwhile, we observe that triCL shows significant improvements on some tasks, e.g., triCL improves the finetune accuracy of SCL by 1.7% on CIFAR-100. Compared to the original contrastive learning methods, the representations learned by triCL keep comparable transferability with stronger identifiability and interpretability.

# 7 Conclusion

In this paper, we propose a new self-supervised paradigm: tri-factor contrastive learning (triCL) which replaces the traditional 2-factor contrast with a 3-factor form $z_x^\top S z_x'$ when calculating the similarity in contrastive learning, where $S$ is a learnable diagonal matrix named importance matrix. With the importance matrix $S$, triCL enables the exact feature identifiability. Meanwhile, the diagonal values in the importance matrix reflect the importance of different features learned by triCL, which means triCL obtains the ordered representations. Moreover, we theoretically prove that the generalization performance of triCL is guaranteed. Empirically, we verify that triCL achieves the feature identifiability, automatically discovers the feature importance and achieves the comparable transferability as current contrastive learning methods.

## Acknowledgements

Yisen Wang was supported by National Key R&D Program of China (2022ZD0160304), National Natural Science Foundation of China (62006153, 62376010, 92370129), Open Research Projects of Zhejiang Lab (No. 2022RC0AB05), and Beijing Nova Program (20230484344).

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

# A  Proofs

## A.1  Proof of Theorem 4.5

*Proof.* We first rewrite $\mathcal{L}_{\text{tri}}$ as a matrix decomposition objective

$$
\begin{aligned}
\mathcal{L}_{\text{tri}}(f) &= -2\mathbb{E}_{x,x^+} f(x)^\top S f(x^+) + \mathbb{E}_x \mathbb{E}_{x^-} \left( f(x)^\top S f(x^-) \right)^2 \\
&= \sum_{x,x'} \left( \frac{A_{xx'}^2}{D_{xx} D_{x'x'}} + D_{xx} D_{x'x'} \left( f(x)^\top S f(x') \right)^2 - 2 A_{xx'} f(x)^\top S f(x') \right) + const \quad (18) \\
&= \|\bar{A} - F S F^\top\|^2.
\end{aligned}
$$

According to the Eckart-Young Theorem [12], the optimal solutions $F^\star, S^\star$ satisfy

$$
F^\star S^\star (F^\star)^\top = U^k \Sigma (V^k)^\top,
$$

where $\Sigma \in \mathbb{R}^{k \times k}$ is a diagonal matrix with the $k$-largest eigenvalues of $\bar{A}$ and $U \in \mathbb{R}^{N \times k}$ contains the corresponding eigenvectors of the $k$-largest eigenvalues. When the regularizer $\mathcal{L}_{\text{Dec}}$ is minimized, $F^\star$ satisfy $(F^\star)^\top F^\star = I$. In the next step, we prove the uniqueness of the optimal solution.

We denote $H = F^\star \Sigma (F^\star)^\top$. As $(F^\star)^\top F^\star = I$, we obtain $H H^\top = F^\star S^\star (S^\star)^\top (F^\star)^\top$. If $\zeta, \sigma$ are a pair of eigenvector and eigenvalue of $H H^\top$, we have

$$
\begin{aligned}
H H^\top \zeta &= F^\star S^\star (S^\star)^\top (F^\star)^\top \zeta = \sigma \zeta, \\
S^\star (S^\star)^\top (F^\star)^\top \zeta &= \sigma (F^\star)^\top \zeta, \qquad\qquad (19) \\
S^\star (S^\star)^\top \left( (F^\star)^\top \zeta \right) &= \sigma \left( (F^\star)^\top \zeta \right).
\end{aligned}
$$

So the eigenvalues of $H H^\top$ are the eigenvalues of $S^\star (S^\star)^\top$. As the positive eigenvalues of $H H^\top$ are uniquely determined and $S^\star$ has a descending order, $S^\star$ is also determined and $S^\star = \sum$.

We note that $H H^\top = F^\star S^\star (S^\star)^\top (F^\star)^\top$, i.e., $H H^\top F^\star = F^\star S^\star (S^\star)^\top$, which means that the $k$ columns of $F^\star$ are the eigenvectors of $H H^\top$ and the corresponding eigenvalues are $\sigma_1 \cdots \sigma_k$. As $H H^\top$ only has $k$ different non-negative eigenvalues $\sigma_1, \cdots, \sigma_k$, the eigenspace of each eigenvalue is one-dimensional. When we consider the real number space, any two eigenvectors $\zeta_i, \zeta_i'$ of the same eigenvalue $\sigma_i$ satisfy $\zeta_i = c \zeta_i'$. As $(F^\star)^\top F^\star = I$, we obtain $c = \pm 1$. As $f(x) = \frac{1}{\sqrt{D_{xx}}} F_x$, we obtain

$$
f_j^\star(x) = \pm \frac{1}{\sqrt{D_{xx}}} \left( U_x^k \right)_j, \quad S^* = \text{diag}(\sigma_1, \ldots, \sigma_k), \qquad (20)
$$

$\square$

## A.2  Proof of Theorem 5.1

*Proof.* The following analysis mainly follows the proofs in [37].

We denote $d_x$ as the $x$-th row of $D$. And we denote $F_m$ as the matrix composed of encoder features on selected dimenstions, *i.e.,* $(F_m)_x = \sqrt{d_x} f^{(m)}(x)$. Recall that $A_{x,x^+} = \mathcal{A}(x, x^+) = \mathbb{E}_{\bar{x} \sim \mathcal{P}_u} \left[ \mathcal{A}(x|\bar{x}) \mathcal{A}(x^+|\bar{x}) \right]$ and $\bar{A}$ is the normalized form of $A$, *i.e.,* $\bar{A}_{x,x^+} = \frac{\mathcal{A}(x,x^+)}{\sqrt{d_x \cdot d_{x^+}}}$. Then we reformulate the downstream error,

$$
\begin{aligned}
\mathbb{E}_{x,y} \|y - W_f f^{(m)}(x)\|^2 &= \sum_{(x,y_x)} d_x \|y_x - W_f f^{(m)}(x)\|^2 \\
&= \|D^{1/2} Y - F_m W_f\|^2 \\
&= \|D^{1/2} Y - \bar{A} C + \bar{A} C - F_m W_f\|^2,
\end{aligned}
$$

where $C_{x,j} = \sqrt{(d_i) \mathbb{1}_{y_x = j}}$. Then we consider the relationship between the downstream error and the augmentation graph, we element-wise consider the matrix $(D^{1/2} Y - \bar{A} C)$,

$$
(D^{1/2} Y)_{x,j} = \sqrt{(d_x) \mathbb{1}_{y_x = j}}, \quad (\bar{A} C)_{x,j} = \sum_{x^+} \frac{A_{x,x^+}}{\sqrt{d_x} \cdot \sqrt{d_{x^+}}} \sqrt{(d_{x^+}) \mathbb{1}_{y_{x^+} = j}}. \qquad (21)
$$

So when $j = y_x$,

$$
\begin{aligned}
(D^{1/2}Y - \bar{A}C)_{x,j} &= \sqrt{(d_x)\mathbb{1}_{y_x=j}} - \sum_{x^+} \frac{\mathcal{A}(x,x^+)}{\sqrt{d_x}}\mathbb{1}_{y_{x^+}=j} \\
&= \sqrt{d_x} - \sum_{x^+} \frac{\mathcal{A}(x,x^+)}{\sqrt{d_x}}\mathbb{1}_{y_{x^+}=j} \\
&= \sum_{x^+} \frac{\mathcal{A}(x,x^+)}{\sqrt{d_x}} - \sum_{x^+} \frac{\mathcal{A}(x,x^+)}{\sqrt{d_x}}\mathbb{1}_{y_{x^+}=j} \\
&= \sum_{x^+} \frac{\mathcal{A}(x,x^+)}{\sqrt{d_x}}\mathbb{1}_{y_{x^+}\neq j} \\
&= \sum_{x^+} \frac{\mathcal{A}(x,x^+)}{\sqrt{d_x}}\mathbb{1}_{y_{x^+}\neq y_x}.
\end{aligned}
\tag{22}
$$

When $j \neq y_x$,

$$
\begin{aligned}
(D^{1/2}Y - \bar{A}C)_{x,j} &= \sqrt{(d_x)\mathbb{1}_{y_x=j}} - \sum_{x^+} \frac{\mathcal{A}(x,x^+)}{\sqrt{d_x}}\mathbb{1}_{y_{x^+}=j} \\
&= 0 - \sum_{x^+} \frac{\mathcal{A}(x,x^+)}{\sqrt{d_x}}\mathbb{1}_{y_{x^+}=j} \\
&= -\sum_{x^+} \frac{\mathcal{A}(x,x^+)}{\sqrt{d_x}}\mathbb{1}_{y_{x^+}=j}.
\end{aligned}
\tag{23}
$$

We define $\beta_x = \sum_{x^+} \mathcal{A}(x,x^+)\mathbb{1}_{y_{x^+}\neq y_x}$, and we have

$$
\begin{aligned}
\|(D^{1/2}Y - \bar{A}C)_x\|^2 &= (\sum_{x^+} \frac{\mathcal{A}(x,x^+)}{\sqrt{d_x}}\mathbb{1}_{y_{x^+}\neq y_x})^2 + \sum_{j\neq y_x}(\sum_{x^+} \frac{\mathcal{A}(x,x^+)}{\sqrt{d_x}}\mathbb{1}_{y_{x^+}=j})^2 \\
&\leq (\sum_{x^+} \frac{\mathcal{A}(x,x^+)}{\sqrt{d_x}}\mathbb{1}_{y_{x^+}\neq y_x})^2 + (\sum_{j\neq y_x}\sum_{x^+} \frac{\mathcal{A}(x,x^+)}{\sqrt{d_x}}\mathbb{1}_{y_{x^+}=j})^2 \\
&\leq (\sum_{x^+} \frac{\mathcal{A}(x,x^+)}{\sqrt{d_x}}\mathbb{1}_{y_{x^+}\neq y_x})^2 + (\sum_{x^+} \frac{\mathcal{A}(x,x^+)}{\sqrt{d_x}}\sum_{j\neq y_x}\mathbb{1}_{y_{x^+}=j})^2 \\
&= (\sum_{x^+} \frac{\mathcal{A}(x,x^+)}{\sqrt{d_x}}\mathbb{1}_{y_{x^+}\neq y_x})^2 + (\sum_{x^+} \frac{\mathcal{A}(x,x^+)}{\sqrt{d_x}}\mathbb{1}_{y_{x^+}\neq y_x})^2 \\
&= \frac{2\beta_x^2}{d_x}.
\end{aligned}
\tag{24}
$$

With that, we obtain $\|D^{1/2}Y - \bar{A}C\| = \sum_x \frac{2\beta_x^2}{d_x}$. As we assume that $E_{\bar{x}\sim\mathcal{P}_u}(\mathcal{A}(x|\bar{x})\mathbb{1}_{y_x\neq\bar{y}}) = \alpha$, so

$$
\begin{aligned}
\sum_{x,x^+} \mathcal{A}(x,x^+)\mathbb{1}_{y_{x^+}\neq y_x} &= \sum_{x,x^+} E_{\bar{x}}(\mathcal{A}(x|\bar{x})\mathcal{A}(x_{x^+}|\bar{x})\mathbb{1}_{y_{x^+}\neq y_x}) \\
&\leq \sum_{x,x^+} E_{\bar{x}}(\mathcal{A}(x|\bar{x})\mathcal{A}(x_{x^+}|\bar{x})(\mathbb{1}_{y_i\neq\bar{y}} + \mathbb{1}_{y_{x^+}\neq\bar{y}})) \\
&= 2E_{\bar{x}\sim\mathcal{P}_d}(\mathcal{A}(x|\bar{x})\mathbb{1}_{y_x\neq\bar{y}}) \\
&= 2\alpha.
\end{aligned}
\tag{25}
$$

Then we have

$$
\begin{aligned}
\|D^{1/2}Y - \bar{A}C\| &= \sum_x \frac{2\beta_x^2}{d_x} \\
&\leq \sum_x 2\beta_x && (d_x = \sum_{x^+} \mathcal{A}(x, x^+) \geq \sum_{x^+} \mathcal{A}(x, x^+)\mathbb{1}_{y_x \neq y_{x^+}}) \\
&= 2\sum_{x,x^+} \mathcal{A}(x, x^+)\mathbb{1}_{y_x \neq y_{x^+}} && (\text{definition of } \beta_x) \\
&\leq 4\alpha. && (\text{Equation (25)})
\end{aligned}
$$

Then we obtain

$$
\begin{aligned}
&\mathbb{E}_{x,y}\|y - W_f f^{(m)}(x)\|^2 \\
&= \|D^{1/2}Y - \bar{A}C + \bar{A}C - F_m W_f\|^2 \\
&\leq 2\|\bar{A}C - F_m W_f\|^2 + 8\alpha && (\|A + B\|^2 \leq 2\|A\|^2 + 2\|B^2\|) \\
&= 2\|(\bar{A} - F_m F_m^T + UU^T)C - F_m W_f\|^2 + 8\alpha \\
&= 2\|(\bar{A} - F_m F_m^T)C + F_m(F_m^T C - W_f)\|^2 + 8\alpha \\
&\leq 4(\|(\bar{A} - F_m F_m^T)C\|^2 + \|F_m(F_m^T C - W_f)\|)^2 + 8\alpha && (\|A + B\|^2 \leq 2(\|A\|^2 + \|B\|^2)) \\
&\leq 4(\|(\bar{A} - F_m F_m^T)\|^2\|C\|^2 + \|F_m\|^2\|(F_m^T C - W_f)\|^2) + 8\alpha && (\|AB\| \leq \|A\|\|B\|) \\
&\leq 4(\|(\bar{A} - F_m F_m^T)\|^2 + 8\alpha && (\|C\| = 1).
\end{aligned}
$$
(26)

In the next step, we analyze the prediction error. We denote $\bar{y}$ as the ground-truth label of original data $\bar{x}$. We first define an ensembled linear predictor $p_f'$. For an original sample, the predictor ensembles the results of all different views and chooses the label predicted the most. With the definition, $\bar{y} \neq p_f'(\bar{x})$ only happens when more than half of the views predict wrong labels. So

$$
\begin{aligned}
\Pr(\bar{y} \neq p_f'(\bar{x})) &\leq 2\Pr(\bar{y} \neq p_f(x)) \\
&\leq 4\mathbb{E}_{\bar{x}\sim\mathcal{P}_d(x), x\sim\mathcal{M}_1(x|\bar{x})}\|\bar{y} - W_f f^{(m)}(x)\|^2 \\
&\leq 8(\mathbb{E}_{x,y}\|y - W_f f^{(m)}(x)\|^2 + \mathbb{E}_{\bar{x}\sim\mathcal{P}_d(x), x\sim\mathcal{M}_1(x|\bar{x})}\|y - \bar{y}\|^2) \\
&\leq 8(\mathbb{E}_{x,y}\|y - W_f f(x)\|^2 + 2\alpha) \\
&\leq 32\|(\bar{A} - F_m F_m^T)\|^2 + 64\alpha + 16\alpha.
\end{aligned}
$$

Then we plug the optimal solutions of SCL (Lemma 4.1) in to $F_m$, and we obtain

$$
\mathcal{E}((f_{SCL}^{\star(m)})) \leq 32((1 - \frac{m}{k})\sum_{i=1}^k \sigma_{l_i}^2 + \sum_{i=k+1}^N \sigma_i^2) + 80\alpha. \tag{27}
$$

When we plug the optimal solutions of triCL (Theorem 4.5) into $F_m$, we obtain:

$$
\mathcal{E}((f_{triCL}^{\star(m)})) \leq 32\sum_{i=m+1}^N \sigma_i^2 + 80\alpha. \tag{28}
$$

$\square$

## A.3 Feature Identifiability of Asymmetric Tri-Factor Contrastive Learning

We first extend the augmentation graph to an asymmetric form. The asymmetric augmentation graph is defined over the set of all samples with its adjacent matrix denoted by $P_O$. In the augmentation graph, each node corresponds to a sample, and the weight of the edge connecting two nodes $x_A$ and $x_B$ is equal to the probability that they are selected as a positive pair, i.e.,$(P_O)_{x_a, x_b} = \mathcal{P}_O(x_a, x_b)$. And we denote $\bar{P}_O$ as the normalized adjacent matrix of the augmentation graph, i.e., $(\bar{P}_O)_{x_a, x_b} = \frac{\mathcal{P}_O(x_a, x_b)^2}{\mathcal{P}_A(x_a)\mathcal{P}_B(x_b)}$.

Similar to the symmetric form, we then rewrite $\mathcal{L}_{\text{tri}}$ as a matrix decomposition objective

$$\mathcal{L}_{\text{tri}}(f_A, f_B, S) = -2\mathbb{E}_{x_a, x_b} f_A(x_a)^\top S f_B(x_b) + \mathbb{E}_{x_a^-, x_b^-} \left(f_A(x_a^-)^\top S f_B(x_b^-)\right)^2$$

$$= \sum_{x_a, x_b} \left(\frac{\mathcal{P}_O(x_a, x_b)^2}{\mathcal{P}_A(x_a)\mathcal{P}_B(x_b)} + \mathcal{P}_A(x_a)\mathcal{P}_B(x_b)\left(f_A(x_a)^\top S f_B(x_L)\right)^2\right.$$

$$\left. - 2\mathcal{P}_O(x_a, x_b) f_A(x_a)^\top S f_B(x_L)\right) + const$$

$$= \|\bar{P}_O - F_A S F_B^\top\|^2.$$

According to the Eckart-Young Theorem [12], the optimal solutions $F_A^\star, S^\star, F_B^\star$ satisfy

$$F_A^\star S^\star (F_B^\star)^\top = U^k \Sigma (V^k)^\top,$$

where $\Sigma \in \mathbb{R}^{k \times k}$ is a diagonal matrix with the $k$-largest eigenvalues of $\bar{P}_O$ and $U \in \mathbb{R}^{N_A \times k}$ contains the corresponding eigenvectors of the $k$-largest eigenvalues. When the regularizer $\mathcal{L}_{\text{Dec}}$ is minimized, $F_A^\star$ and $F_B^\star$ satisfy $(F_A^\star)^\top F_A^\star = I, (F_B^\star)^\top F_B^\star = I$. In the next step, we prove the uniqueness of the optimal solution.

We denote $H = F_A^\star \Sigma F_B^\star$, and we obtain $HH^\top = F_A^\star S^\star (S^\star)^\top (F_A^\star)^\top$. If $\zeta, \sigma$ are a pair of eigenvector and eigenvalue of $HH^\top$, we have

$$HH^\top \zeta = F_A^\star S^\star (S^\star)^\top (F_A^\star)^\top \zeta = \sigma \zeta,$$

$$S^\star (S^\star)^\top (F_A^\star)^\top \zeta = \sigma (F_A^\star)^\top \zeta, \tag{29}$$

$$S^\star (S^\star)^\top \left((F_A^\star)^\top \zeta\right) = \sigma \left((F_A^\star)^\top \zeta\right).$$

So the eigenvalues of $HH^\top$ are the eigenvalues of $S^\star (S^\star)^\top$. As the positive eigenvalues of $HH^\top$ are uniquely determined and $S^\star$ has an increasing order, $S^\star$ is also determined and $S^\star = \sum$.

We note that $HH^\top = F_A^\star S^\star (S^\star)^\top (F_A^\star)^\top$, i.e., $HH^\top F_A^\star = F_A^\star S^\star (S^\star)^\top$, which means that the $k$ columns of $F_A^\star$ are the eigenvectors of $HH^\top$ and the corresponding eigenvalues are $\sigma_1 \cdots \sigma_k$. As $HH^\top$ only has $k$ different non-negative eigenvalues $\sigma_1, \cdots, \sigma_k$, the eigenspace of each eigenvalue is one-dimensional. When we consider the real number space, any two eigenvectors $\zeta_i, \zeta_i'$ of the same eigenvalue $\sigma_i$ satisfy $\zeta_i = c\zeta_i'$. As $(F_A^\star)^\top F_A^\star = I$, we obtain $c = +1$. Then we eliminate the ambiguity of the sign following Eq 11 and $F_A^\star$ is unique. Similarly, $F_B^\star$ is also unique. So the optimal solution of $\mathcal{L}_{\text{triCLIP}}$ is unique.

## B    Experimental Details

### B.1    Experiment Details of Section 6.1

We first generate a random matrix $A$ with size $5000 \times 3000$, and make sure that it does not contain multiple eigenvectors (which is easy to satisfy). For the matrix factorization problem $\|A - FG^\top\|_F^2$, we apply off-the-shelf algorithms and repeat this process ten times. We then calculate the mean and variance of the $l_2$ pairwise distance between the obtained solutions of $F$. For the trifactorization objective $\|A - FSG^\top\|_F^2$, we use SVD to obtain an initial solution and apply the sign identification procedure to determine the sign of each eigenvector. Similarly, we also repeat this process ten times and calculate the mean and variance of the $l_2$ pairwise distance between different solutions.

### B.2    Experiment Details of Section 6.2

**Pretraining Setups.** For different evaluation tasks (k-NN, linear evaluation, image retrieval), we use the same pretrained models. We adopt ResNet-18 as the backbone. For CIFAR-10 and CIFAR-100, the projector is a two-layer MLP with hidden dimension 2048 and output dimension 256. And for ImageNet-100, the projector is a two-layer MLP with hidden dimension 4096 and output dimension 512. We pretrain the models with batch size 256 and weight decay 0.0001. For CIFAR-10 and CIFAR-100, we pretrain the models for 200 epochs. While for ImageNet-100, we pretrain the models for 400 epochs. We use the cosine anneal learning rate scheduler and set the initial learning rate to 0.4 on CIFAR-10, CIFAR-100, and 0.3 on ImageNet-100.

As the importance matrix is learned on the projection layer, we conduct the downstream tasks on the features encoded by the complete networks (containing both the backbones and the projectors).

**The Distribution of the Importance Matrix.** When observing the distribution of feature importance discovered by the importance matrix $S$, we first apply the softplus activation functions on the diagonal values of S and sort different rows of $S$ by the descending order of corresponding diagonal values in $S$. We denote the non-negative ordered diagonal values of $S$ as $(s_1, \cdots, s_k)$. When we present the distribution of them in Figure 2(a), we normalize the diagonal values and obtain $(s_1/\sum_{i=1}^{k} s_i, \cdots, s_k/\sum_{i=1}^{k} s_i)$.

**The K-NN Accuracy on Selected Dimensions.** For k-NN evaluation on 10 selected dimensions, we do not finetune the models. We sort the dimensions of $f(x)$ by the descending order of corresponding diagonal values in the importance matrix. The k-NN is conducted on the standard split of CIFAR-10, CIFAR-100 and ImageNet-100 and the predicted label of samples is decided by the 10 nearest neighbors.

**Linear Evaluation on Selected Dimensions.** We train the linear classifier on 20 dimensions of the frozen networks for 30 epochs during the linear evaluation. We set batch size to 256 and weight decay to 0.0001. For triCL, we sort the dimensions by descending order of the importance matrix. And for SCL, we randomly choose 20 dimensions.

## C More Extensions of Tri-factor Contrastive Learning

In this section, we apply tri-factor contrastive learning to another representative contrastive learning objective: the non-contrastive loss [15, 5].

Besides contrastive learning, non-contrastive learning is another popular self-supervised framework that throws the negative samples in contrastive learning and learns the meaningful representations only by aligning the positive pairs. Taking the state-of-the-art algorithm BYOL [15] as an example, they use an MSE loss:

$$\mathcal{L}_{\text{MSE}}(f, g) = 2 - 2 \cdot \mathbb{E}_{x.x^+} \frac{g(x)^\top f(x^+)}{\|g(x)\|_2 \cdot \|f(x^+)\|_2}, \tag{30}$$

where $g(x)$ and $f(x)$ are two different networks to avoid the feature collapse. Then we consider adapting the tri-term loss to the non-contrastive learning, i.e.,

$$\mathcal{L}_{\text{triMSE}}(f, g) = 2 - 2 \cdot \mathbb{E}_{x.x^+} \frac{g(x)^\top S f(x^+)}{\|g(x)\|_2 \cdot \|f(x^+)\|_2} + \left\|\mathbb{E}_x g(x) g(x)^\top - I\right\|^2. \tag{31}$$

It is noticed that BYOL utilizes the stop-gradient technique on the target network $f$ and it is updated by exponential moving average. So we only calculate the feature decorrelation loss on the online network $g$.

