## A Proofs

### A.1 Proof of Theorem 4.5

*Proof.* We first rewrite $\mathcal{L}_{\mathrm{tri}}$ as a matrix decomposition objective

$$
\begin{aligned}
\mathcal{L}_{\mathrm{tri}}(f) &= -2\mathbb{E}_{x,x^+} f(x)^\top S f(x^+) + \mathbb{E}_x \mathbb{E}_{x^-} \left( f(x)^\top S f(x^-) \right)^2 \\
&= \sum_{x,x'} \left( \frac{A_{xx'}^2}{D_{xx}D_{x'x'}} + D_{xx}D_{x'x'} \left( f(x)^\top S f(x') \right)^2 - 2A_{xx'} f(x)^\top S f(x') \right) + const \quad (16) \\
&= \| \bar{A} - FSF^\top \|^2.
\end{aligned}
$$

According to the Eckart-Young Theorem [Eckart and Young, 1936], the optimal solutions $F^\star, S^\star$ satisfy

$$
F^\star S^\star (F^\star)^\top = U^k \Sigma (V^k)^\top,
$$

where $\Sigma \in \mathbb{R}^{k \times k}$ is a diagonal matrix with the $k$-largest eigenvalues of $\bar{A}$ and $U \in \mathbb{R}^{N \times k}$ contains the corresponding eigenvectors of the $k$-largest eigenvalues. When the regularizer $\mathcal{L}_{\mathrm{Dec}}$ is minimized, $F^\star$ satisfy $(F^\star)^\top F^\star = I$. In the next step, we prove the uniqueness of the optimal solution.

We denote $H = F^\star \Sigma (F^\star)^\top$. As $(F^\star)^\top F^\star = I$, we obtain $HH^\top = F^\star S^\star (S^\star)^\top (F^\star)^\top$. If $\zeta, \sigma$ are a pair of eigenvector and eigenvalue of $HH^\top$, we have

$$
\begin{aligned}
HH^\top \zeta &= F^\star S^\star (S^\star)^\top (F^\star)^\top \zeta = \sigma \zeta, \\
S^\star (S^\star)^\top (F^\star)^\top \zeta &= \sigma (F^\star)^\top \zeta, \quad\quad\quad (17) \\
S^\star (S^\star)^\top \left( (F^\star)^\top \zeta \right) &= \sigma \left( (F^\star)^\top \zeta \right).
\end{aligned}
$$

So the eigenvalues of $HH^\top$ are the eigenvalues of $S^\star (S^\star)^\top$. As the positive eigenvalues of $HH^\top$ are uniquely determined and $S^\star$ has a descending order, $S^\star$ is also determined and $S^\star = \sum$.

We note that $HH^\top = F^\star S^\star (S^\star)^\top (F^\star)^\top$, i.e., $HH^\top F^\star = F^\star S^\star (S^\star)^\top$, which means that the $k$ columns of $F^\star$ are the eigenvectors of $HH^\top$ and the corresponding eigenvalues are $\sigma_1 \cdots \sigma_k$. As $HH^\top$ only has $k$ different non-negative eigenvalues $\sigma_1, \cdots, \sigma_k$, the eigenspace of each eigenvalue is one-dimensional. When we consider the real number space, any two eigenvectors $\zeta_i, \zeta_i'$ of the same eigenvalue $\sigma_i$ satisfy $\zeta_i = c\zeta_i'$. As $(F^\star)^\top F^\star = I$, we obtain $c = \pm 1$. As $f(x) = \frac{1}{\sqrt{D_{xx}}} F_x$, we obtain

$$
f_j^\star(x) = \pm \frac{1}{\sqrt{D_{xx}}} \left( U_x^k \right)_j, \, S^* = \mathrm{diag}(\sigma_1, \dots, \sigma_k), \quad\quad\quad (18)
$$

$\square$

### A.2 Proof of Theorem 5.1

We first introduce a lemma which theoretically guarantees the generalization performance of spectral contrastive learning.

**Lemma A.1** ([HaoChen et al., 2021]). *For the optimal solutions to spectral contrastive learning (SCL), we have*

$$
\mathcal{E}(f_{SCL}^\star) \leq \mathcal{O}(\frac{\alpha}{1 - \sigma_{k+1}}),
$$

*where we denote $\alpha$ as the probability that the natural samples and augmented views have different labels, i.e., $\alpha = \mathbb{E}_{\bar{x} \sim \mathcal{P}_u} \mathbb{E}_{x \sim \mathcal{A}(\cdot | \bar{x})} \mathbb{1}[y(\bar{x}) \neq y(x)]$ and $\sigma_{k+1}$ as the $(k+1)$-th largest eigenvalue of the normalized adjacent matrix $\bar{A}$.*

Then we construct the generalization guarantee of tri-contrastive learning.

*Proof.* Following the proof of Theorem 4.5, we know that the optimal solutions learned by triCL are

$$
\begin{aligned}
F^\star &= U^k, \\
S^\star &= diag(\sigma_1, \cdots \sigma_k).
\end{aligned}
$$

450 So we know that the optimal encoder of triCL satisfies, $\forall x \in \mathcal{D}$

$$f^*(x) = \frac{1}{\sqrt{D_{xx}}} \left(U_x^k\right)^\top.$$

451 Compared with the optimal solutions of spectral contrastive learning (Eq. 2), we know

$$(\text{diag}(\sigma_1, \cdots \sigma_k)R)^\top f^*_{triCL}(x) = f^\star_{SCL}(x), \tag{19}$$

452 where $f^\star_{triCL}, f^\star_{SCL}$ denote the optimal solutions of tri-contrastive learning and spectral contrastive
453 learning. As $\text{diag}(\sigma_1, \cdots \sigma_k)R$ is an invertible matrix, we then prove that the invertible matrix can
454 be absorbed in the linear probing. We denote $\text{diag}(\sigma_1, \cdots \sigma_k)R$ as Q and we denote the linear
455 classifier as $B$, i.e., $g(f(x)) = f(x)^\top B$. For a linear classfier $B$, let $\tilde{B} = BQ^{-1}$. We then obtain
456 $f^\star_{triCL}(x)^\top \tilde{B} = f^\star_{SCL}(x)^\top B$.

457 So

$$\mathcal{E}(f^\star_{triCL}) = \mathcal{E}(f^\star_{SCL}).$$

458 With lemma A.1, we have

$$\mathcal{E}(f^\star_{triCL}) \leq \mathcal{O}(\frac{\alpha}{1 - \sigma_{k+1}}).$$

459 $\qquad\qquad\qquad\qquad\qquad\qquad\qquad\qquad\qquad\qquad\qquad\qquad\qquad\qquad\qquad\qquad\qquad$ $\square$

## A.3 Proof of Theorem 5.2

461 *Proof.* Based on the proof of Theorem 4.5, we know that the $t$-th dimension of the optimal solutions
462 satisfies

$$F^\star = U_t^k,$$
$$S_t^\star = diag(\sigma_1, \cdots \sigma_k)_t.$$

463 With the analysis in Eckart-Young theorem [Eckart and Young, 1936], we have

$$\|\bar{A} - F_t^\star S_t^\star (F_t^\star)^\top\|_F^2 = \|\bar{A} - U_t^k \, \text{diag}(\sigma_1, \cdots \sigma_k)_t (U_t^k)^\top\|_F^2$$
$$= \sum_{i=1}^{t-1} \sigma_i^2 + \sum_{i=t+1}^{k} \sigma_i^2.$$

464 As $\sigma_i$ is the $i$-th largest eigenvalues of $\bar{A}$, so

$$\|\bar{A} - F_1^\star S_1^\star (F_1^\star)^\top\|_F^2 \leq \cdots \leq \|\bar{A} - F_k^\star S_k^\star (F_k^\star)^\top\|_F^2.$$

465 Following Eq 16, we obtain

$$\mathcal{L}_{triCL}(f_t, S_t) = \|\bar{A} - F_t^\star S_t^\star (F_t^\star)^\top\|_F^2 + const,$$

466 we obtain

$$\mathcal{L}_{\text{triCL}}(f_1^\star, S_1^\star) \leq \cdots \leq \mathcal{L}_{\text{triCL}}(f_k^\star, S_k^\star).$$

467 $\qquad\qquad\qquad\qquad\qquad\qquad\qquad\qquad\qquad\qquad\qquad\qquad\qquad\qquad\qquad\qquad\qquad$ $\square$