# OpenReview forum: "Identifiable Contrastive Learning with Automatic Feature Importance Discovery"
_NeurIPS.cc/2023/Conference — NeurIPS 2023 poster_

### Official Review · Reviewer_aafx · 2023-06-23

**Soundness:** 2 fair
**Presentation:** 3 good
**Contribution:** 2 fair
**Rating:** 4
**Confidence:** 4

**Summary:**

In this paper, the authors proposed a new contrastive learning scheme (tri-contrastive learning) with a learnable diagonal matrix, which can capture the feature importance. To show the significance of the proposed method, the authors conducted both theoretical and experimental analysis, where some interesting findings are observed. However, it is not clear how the proposed method advances existing contrastive learning-based methods, and experimental results are not very convincing. The detailed comments are below.

**Strengths:**

The idea is simple and interesting, with reasonable theoretical analysis based on eigenspace.



**Weaknesses:**

1. The idea of analyzing feature importance for contrastive learning is not new [a]. While the proposed method is different from [a], the submission lacks analysis how the proposed method outperforms [a], especially based on the generalization capability the submission focused on.

[a] Accurate and Robust Feature Importance Estimation under Distribution Shifts, aaai21

2. The visualization example in fig 1 is not clear. Noted that the contrastive learning is performing on the feature space, it is not clear to me based on the paper how the feature space linked to the image space.

3. The generalization bound is not very clear, the upper bound depends on the term \alpha, which relies on data augmentation. How different augmentations inclunce the term \alpha. In what condition that the sample and its augmented view could have different labels?

4. The improvement on three benchmark datasets are very marginal (almost no improvement), and the results can’t justify the generalization capability claimed in this paper, convergence analysis is also not presented.

**Questions:**

Please see above.

---

> ### Author Rebuttal · Authors · 2023-08-09
>
> We thank reviewer aafx for appreciating our work on the novelty and simplicity. We will address the main weaknesses in the following.
>
> ---
>
> Q1: The comparison between [a] and tri-contrastive learning.
>
> A1: Following your suggestions. we compare ProFILE proposed in [a] with triCL from different aspects. First, they are quite different in methodology:
>
> - **Different levels of feature importance.** We note that PRoFILE proposed by [a] estimates the importance of different dimensions of **input samples** while tri-contrastive learning obtains the importance order of **encoded features.**
> - **Different theoretical guarantees.** We note that PRoFILE designs an empirical objective to estimate the importance of different dimensions, and they do not theoretically analyze the generalization performance. In contrast, triCL establish the theoretical bounds in Section 5.1 and 5.2.
>
> Empirically, we compare the generalization performance between triCL and PRoFILE.
>
> *Table A: Comparison between SCL, PRoFILE, and triCL in linear evaluation with selected dimensions on CIFAR-10.*
>
> |  | SCL (Random 20 dimensions) | PRoFILE (20 dimensions with largest estimated importance) | triCL (20 dimensions with largest S values) |
> | --- | --- | --- | --- |
> | Linear Accuracy  | 69.0 | 74.3 | **82.0** |
>
> As shown in the above Table, although PRoFILE shows better performance than the random selections, triCL still achieves significant improvements over PRoFILE (82.0 vs 74.3). In summary, triCL outperforms PRoFIle both theoretically and empirically.
>
> [a] Accurate and Robust Feature Importance Estimation under Distribution Shifts, AAAI21.
>
> ---
>
> Q2: The visualization example in Fig 1 is not clear. Noted that the contrastive learning is performing on the feature space, it is not clear based on the paper how the feature space linked to the image space.
>
> A2: In Figure 1, we visualize the features learned by triCL by presenting them with the corresponding inputs before being encoded. We add Figure B in rebuttal pdf to better explain the visualization process. In practice, we first **encode the images** by the encoders pretrained with tri-contrastive loss (with importance matrix $S$) to get their features.  Then we respectively select the 10 most and least important dimensions in the feature according to the importance matrix $S$ of the encoder. We present **each selected dimension by each row filled by images that have the largest activation on this dimension**. As shown in Figure 1, the images whose features are activated in the dimensions associated with the highest values in the importance matrix consistently belong to the same class, while the samples activated in the dimensions with the lowest values in S appear to be mostly independent, which further confirms that the importance matrix effectively reflects the significance of different feature dimensions.
>
> ---
>
> Q3: In what condition that the sample and its augmented view could have different labels? How different augmentations influence the term \alpha?
>
> A3: **When the augmentation is too strong, (e.g., the scale of the crop is too large), the augmented views may lose the semantic information and the labels would be changed.** For example, as shown in Figure C in rebuttal pdf, for an image that contains a car and a person, when the augmentation strength is aggressive, the augmented view may lose the center objective (car) and focus on the person. In this case, it is obvious that the augmented views of the same images would have different labels. And in the extreme case, the augmented views may only focus on the background messages and lose all the semantic information of objectives. Based on the analysis and the examples above, we know that the probability that two augmented views of the same image have different labels is mainly decided by the choices of data augmentation. To be specific, **when we use more aggressive augmentations, there will exist more augmented views that have different labels, which implies a larger labeling error $\alpha$.** So the generalization bound indicates that the triCL needs to choose appropriate augmentation strength and avoid too-aggressive augmentations to obtain better generalization performance.
>
> ---
>
> Q4: The improvements on three benchmark datasets are very marginal, and the results can’t justify the generalization capability, convergence analysis is also not presented.
>
> A4: There may be some misunderstanding here. In fact, by combing Theorem 5.1 and the generalization guarantee of the 2-factor contrastive loss in [b], what we try to prove theoretically is that triCL and the original spectral contrastive learning share the **equal** generalization bound. **So the comparable downstream performance in feature importance irrelevant tasks between two contrastive objectives is expected and exactly verifies the theoretical analysis.** Meanwhile, the main advantages of our proposed methods are the feature identifiability and interpretability.  For example, the experiments in Section 6.2 show that triCL achieves significantly superior k-NN accuracy, linear accuracy, and image retrieval precision on selected dimensions.
>
> Besides, we conduct additional experiments to compare the training process of triCL and the original contrastive learning. To be specific, we compare the training loss of SimCLR and tri-SimCLR on ImageNet-100. As shown in Figure A in rebuttal.pdf, we observe that the two objectives have similar training curves, which implies that these two objectives converge at a similar pace.
>
> [b] Jeff Z HaoChen, Colin Wei, Adrien Gaidon, and Tengyu Ma. Provable guarantees for self-supervised deep learning with spectral contrastive loss. In NeurIPS, 2021.

---

> ### Author Response · Authors · 2023-08-17
> **Your invaluable input is needed**
>
> Dear Reviewer aafx, thanks for your time reviewing our paper. We have meticulously prepared a detailed response addressing the concerns you raised. Could you please have a look to see if there are further questions? Your invaluable input is greatly appreciated. Thank you once again, and we hope you have a wonderful day!

---

> ### Author Response · Authors · 2023-08-20
> **Your further inputs are greatly appreciated! Only one day left.**
>
> Dear Reviewer aafx,
>
> For your raised questions, we prepared a detailed response to address your concerns. We were hoping to hear your feedback on them.
>
> As there is only one day left for the reviewer and author discussions and we understand that everyone has a tight schedule, we kindly wanted to send a gentle reminder to ensure that our response sufficiently addressed your concerns or if there are further aspects we need to clarify.
>
> If you could find the time to provide your thoughts on our response, we would greatly appreciate it.
>
> Best, Authors

---

> > ### Comment · Reviewer_aafx · 2023-08-21
> > **Thanks**
> >
> > Thanks for the rebuttal. I am still not convinced about the part of feature importance, especially the generalization capability. Thus, I will keep my score unchanged.

---

> > > ### Author Response · Authors · 2023-08-22
> > > **Could you let us know what is your remaining concern?**
> > >
> > > Thanks for your reply! However, we do not quite get the reasons why you are still not convinced and hope for your further explanations.
> > >
> > > From our perspective, we have properly addressed you previous concerns in these two aspects. **For the feature importance part**, since ProFILE does not have formal theoretical guarantees, in **A1**, we compare with it empirically and show that our methods obtain significantly better performance over ProFILE (74.3% vs 82.0%).
> > >
> > > **For the generalization capacity part**, in **A4**, we have clarified that the generalization guarantee in Theorem 5.1 is established for **the case when using** **all learned features for linear probing (instead of using a subset of important features).** Thus, it mainly serves as a proof of concept, showing that **in this extreme case, tri-CL matches the generalization error rate of 2-factor CL**. However, since 2-factor CL does not learn feature importance, it can only select random subsets of features while ours can select directly the most important ones based on $S$. This makes our method stand out when using **a subset of important features**.
> > >
> > > Given the explanations above, we are keen to learn what are your remaining concerns on these two aspects. Look forward to your further reply!

---

### Official Review · Reviewer_Zqsm · 2023-07-03

**Soundness:** 3 good
**Presentation:** 3 good
**Contribution:** 3 good
**Rating:** 6
**Confidence:** 3

**Summary:**

The paper proposes a new contrastive triCL method, which extends the existing pair-wise contrastive framework. The core motivation is the inability of CL methods to deliver identifiable representations. To tackle this, the paper proposes factorizing the pair-wise similarity measure x^T@y into a bilinear x^T@S@y form, where learnable matrix S is a diagonal feature importance matrix; and theoretically proves the identifiability property. The paper also demonstrates possible automated feature importance discovery by analyzing the learned feature importance matrix.

**Strengths:**

1. The paper addresses an important problem of identifiability, which is often overlooked in contrastive learning.

2. The proposed method is elegant in its simplicity. Also, due to its simplicity, it does not introduce any major computational overheads.

3. Feature importance discovery can be an important component to speed up the inference (by excluding features of low importance)

**Weaknesses:**

1. In my opinion, the term tri-contrastive (triCL) does incorrectly reflect the essence of the method. The proposed method still is based on pair-wise contrast in a nutshell, although the contrast is in the factorized form.

2. It seems that BarlowTwins tackles almost the same problem as the paper states in L33-34. On the technical side, the L_dec term Eq (8) seems to be a special case of the Barlow twins' objective. In that sense, the L_dec term in triCL can be considered not as just a regularization, but as a main source of the supervisory signal.

3. The ablation is needed to verify the importance of the L_dec term in the triCL objective. I would be interested to see the results for L_dec, L_tri, and L_dec + L_tri objectives separately.

4. Experimental evaluation is limited to relatively small-scale datasets and only two baseline contrastive learning methods.

UPD: The rebuttal addressed most of my concerns.

**Questions:**

1. Why is the softmax in S? Wouldn't the softmax limit the flexibility of each feature dimension, when computing x^T@S@y similarity? Have authors tried other normalization approaches, e.g. sigmoid for each diagonal element of S?

2. Does identifiability of representation always boils down to reproducibility within the scope of this work?

**Limitations:**

I suggest adding a paragraph on limitations, which would highlight the scope of application of the proposed approach (or is the proposed method expected to work better in all circumstances?)

---

> ### Author Rebuttal · Authors · 2023-08-09
>
> We thank reviewer Zqsm for appreciating our work on its importance and simplicity. We will address the weaknesses and questions you mentioned.
>
> ---
>
> Q1： The term tri-contrastive does incorrectly reflect the essence of the method.
>
> A1:  Here, the term “tri-contrastive” does not imply that we change the pair-wise contrast to three encoders. Instead, it refers to the proposed importance matrix $S$. We note that **our method is deeply connected to the matrix tri-factorization approach** [a] that transforms the matrix decomposition objective from $\Vert A - UV \Vert ^2$ to $\Vert A - USV \Vert^2$. Therefore, the name triCL indicates the technique we use here. Alternatively, the name “Contrastive Learning with Importance Discovery (CLID)” could reveal the main focus of this method more clearly. Please let us know if you have better options.
>
> [a] Orthogonal nonnegative matrix tri-factorizations for clustering. KDD 2006.
>
> ---
>
> Q2: It seems that BarlowTwins tackles almost the same problem.
>
> A2: Although the BarlowTwins loss and the $L_{dec}$ term are similar, we note that **the triCL objective consists of two terms: $L_{tri}$ and $L_{dec}$**. It is important to highlight that we **cannot achieve feature identifiability only with $L_{dec}$**  (and thus Barlow Twins).
>
> Specifically, we show a counter-example. If we only consider $L_{dec}$, i.e., $\Vert \mathbb{E}_{x}f(x)f(x)^\top-I\Vert^2,$ and denote $f^\star$ as an optimal solution of it, $R$ as an **arbitrary** orthogonal matrix, then we note that $f^\star R$  is also an optimal solution.
>
> Consequently, **only introducing $L_{dec}$**  **cannot enable the feature identifiability and the existence of $L_{tri}$ term makes triCL and BarlowTwins essentially different.**
>
> ---
>
> Q3: The ablation on the $L_{dec}$  term in triCL.
>
> A3: We conduct additional experiments to verify the importance of both the $L_{tri}$ and $L_{dec}$ terms.
>
> *Table A: The linear accuracy of selected features on CIFAR-10.*
>
> |  | $L_{tri}$  (20 dimensions with largest S values) | $L_{dec}$  (Random 20 dimensions) | $L_{dec}$ + $L_{tri}$ (20 dimensions with largest S values) |
> | --- | --- | --- | --- |
> | Linear Accuracy  | 76.5 | 70.5 | **82.0** |
>
> As shown in the above table, both the $L_{dec}$ and $L_{tri}$ are important in importance-relevant tasks.
>
> ---
>
> Q4: Experimental evaluation on other datasets and methods.
>
> A4: Following your suggestions, we conduct experiments on ImageNet and apply triCL to another baseline BarlowTwins.
>
> *Table B: The comparison in feature importance relevant tasks on ImageNet.*
>
> |  | SimCLR (Random 40 dimensions) | Tri-SimCLR (40 dimensions selected by S) |
> | --- | --- | --- |
> | Linear Accuracy | 28.9 | **34.1** |
> | Retrieval mAP | 15.4 | **22.7** |
>
> *Table C: The linear accuracy of selected features on CIFAR-10.*
>
> |  | BarlowTwins (Random 50 dimensions) | Tri-BarlowTwins (50 dimensions selected by S) |
> | --- | --- | --- |
> | Linear Accuracy | 55.9 | **63.2** |
>
> As shown in the above tables,  triCL achieves significant improvements in the feature importance relevant tasks, which further verifies that triCL can be applied in different datasets and frameworks.
>
> ---
>
> Q5: Why is the softmax in S? Wouldn't it limit the flexibility? other approaches?
>
> A5: Here, Softmax normalizes the values in $S$ to avoid the dramatic change of loss values for numerical stability. Meanwhile, it also implements a global receptive field over all importance values. Like the L2 normalization adopted in SimCLR, softmax sacrifices a bit of flexibility, but from the table below, we observe that it indeed performs the best among other choices (such as Sigmoid and ReLU).
>
> *Table D: The linear accuracy of selected features on CIFAR-10.*
>
> |  | SCL  (Random 20 dimensions) | Sigmoid  (20 dimensions with largest S values) | ReLU (20 dimensions with largest S values) | Softmax (20 dimensions with largest S values) |
> | --- | --- | --- | --- | --- |
> | Linear Accuracy  | 69.0 | 81.1 | 78.4 | **82.0** |
>
> ---
>
> Q6: Does identifiability of representation always boils down to reproducibility within the scope of this work?
>
> A6: In fact, identifiability brings various advantages besides the reproducibility, like interpretability, robustness, and superiority in casual inference [b]. Specifically, we demonstrate two advantages of triCL.
>
> **Feature interpretability and disentanglement.**  Theorem 4.5 shows that with identifiability, the diagonal values of $S$ are the singular values of the augmentation graph, i.e., they are interpretable. Besides, as the identifiability eliminates the freedom of linear transformations in original CL, the features are decoupled.  For example, Figure 1(a) indicates that each dimension in triCL only captures the semantics of one class.
>
> **Better Feature Transferability.** As the identifiability can decouple causal features and non-causal features, the transferability may be improved when facing the distribution shift [c].  Empirically, the following table shows that tri-SimCLR achieves improvements in the transfer learning task.
>
> *Table E: The transfer accuracy from ImageNet to CIFAR-10.*
>
> |  | SimCLR | Tri-SimCLR |
> | --- | --- | --- |
> | Linear Accuracy | 45.6 | **46.5(+0.9)** |
>
> [b] Introductory overview of identifiability analysis: A guide to evaluating whether you have the right type of data for your modeling purpose. In Environmental Modelling & Software, 2019.
>
> [c] Toward causal representation learning. In Proceedings of the IEEE. 2021.
>
> ---
>
> Q7: Limitations.
>
> A7: The primary advantages of our proposed tri-contrastive learning approach are enhanced feature identifiability and interpretability. This method proves to be particularly beneficial for tasks that require identifiable and explainable features, such as image retrieval tasks. However, for importance-relevant tasks (e.g., linear probing and finetuning), triCL only achieves comparable performance as conventional contrastive learning.

---

> ### Author Response · Authors · 2023-08-17
> **Your invaluable input is needed**
>
> Dear Reviewer Zqsm, thanks for your time reviewing our paper. We have meticulously prepared a detailed response addressing the concerns you raised. Could you please have a look to see if there are further questions? Your invaluable input is greatly appreciated. Thank you once again, and we hope you have a wonderful day!

---

> > ### Comment · Reviewer_Zqsm · 2023-08-18
> >
> > In my opinion, "Contrastive Learning with Importance Discovery (CLID)” better reflects the essence of the proposed method. The connection with tri-factorization could be highlighted in the main text.

---

> > > ### Comment · Reviewer_Zqsm · 2023-08-18
> > >
> > > I thank authors for the response. The rebuttal addresses my concerns, given the rebuttal experiments and clarifications are added to the main paper. I thus raise my score.

---

### Official Review · Reviewer_tkty · 2023-07-05

**Soundness:** 2 fair
**Presentation:** 3 good
**Contribution:** 2 fair
**Rating:** 5
**Confidence:** 4

**Summary:**

Traditional CL calculates the similarities between two samples while this paper proposes to insert a learnable diagonal matrix between two feature vectors multiplication and name it triCL. The authors claim and show the proposed method leads to the exact interpretability of learned features. Some theoretical analysis is provided.

**Strengths:**

The paper is well written and the proposed idea is simple and may be useful for explaining the CL.

**Weaknesses:**

1. Lack of novelty: The proposed method is essentially a bilinear function, and it can be rewritten the same as conventional CL, i.e., $f_1^TSf_2 = (\tilde{S}f_1)^T(\tilde{S}f_2)$ where $S=\tilde{S}^T\tilde{S}$ since S is a nonnegative diagonal matrix, i.e., a kernel matrix. From this point, the new parameter $S$ can be integrated into the neural network training, without explicit consideration. $S$ may be useful for the interpretation of CL, but at this point, I do not see how different using $S$ is from using traditional features with some statistical pro-processing such as norm. At least, I do not see the necessity of the proposed method in terms of interpretability.

2. Weak experiments: From Table 1, the improvement is marginal and even worse than the baseline. This is probably because introducing bilinear forms makes the training easier stuck in local minima. Also, these results verify my thoughts about bilinear form in learning above. I suggest the authors to investigate more in training, e.g., plotting training curves and doing some comparisons.

**Questions:**

see my comments above.

**Limitations:**

no discussion on limitations is provided, and i suggest the authors to add this part

---

> ### Author Rebuttal · Authors · 2023-08-09
>
> We thank reviewer tkty for appreciating our work on the interpretability of contrastive learning. We understand your main concerns are on whether we can integrate the importance matrix into the neural network and rewrite tri-contrastive learning the same as original contrastive learning. We will elaborate on this part in detail and address the weaknesses you mentioned.
>
> ---
>
> Q1: The proposed method can be rewritten the same as conventional CL, i.e., $f_1^\top Sf_2 = (\tilde{S}f_1)^\top(\tilde{S}f_2)$, where $S = \tilde{S}^\top S$ since S is a nonnegative diagonal matrix, i.e., a kernel matrix. And the new parameter can be integrated into the neural network training.
>
> A1: Indeed, the importance matrix can be integrated into the 2-branch architecture, but we note that **the triCL objective is essentially different from canonical 2-branch CL objectives and cannot be reduced**.
>
> To see this, remind that the tri-contrastive objective consists of two losses, $L_{tri}(f, S) = 2E_{x,x^+}f(x)^\top S f(x^+) +E_{x}E_{x^-}\left(f(x)^\top S f(x^-)\right)^2$ and $L_{dec}(f) = \left\Vert E_{x}f(x)f(x)^\top-I\right\Vert^2$. By integrating the importance matrix into the neural network ($f' = \tilde{S}f$), the first term ($L_{tri}$) can be rewritten as the original contrastive learning, and the second term ($L_{dec}$) becomes $L_{dec}(f') = \left\Vert E_{x}f'(x)f'(x)^\top-I\right\Vert^2 = \left\Vert E_{x}(\tilde{S}f(x))(\tilde{S}f(x))^\top-I\right\Vert^2$. In this situation,  the optimal solutions $f^\star, S^\star$ are not unique and the feature identifiability will disappear.
>
> To verify it, we use the following counter-example. We denote $f'^\star=\tilde{S}^\star f^\star$ as the optimal solutions of $L_{dec}(f')$ and $R$ as an **arbitrary** invertible matrix. Then $f'^\star = \tilde{S}^\star R R^{-1} f^\star$ are also optimal solutions (the learned encoder is $R^{-1}f^\star$ and the learned importance matrix is $\tilde{S}^\star R$ ) . In this case, it is obvious that the values in the importance matrix will be easily perturbed by the variable $R$ , which means the importance matrix can not indicate the feature importance anymore.  Therefore, we cannot reduce triCL to a canonical CL objective using a single output feature for each branch.
>
> The use of the name triCL is because this idea originates from and has a close connection to the matrix tri-factorization method with a 3-factor objective $\Vert A-USV\Vert^2$ such that $U,V$ are orthogonal matrices [a], and it is clear that this problem cannot be reduced to a 2-factor problem.
>
> [a] Orthogonal nonnegative matrix tri-factorizations for clustering. In *ACM SIGKDD* 2006.
>
> ---
>
> Q2: From Table 1, the improvement is marginal and even worse than the baselines. And there needs more experiments to investigate the training process.
>
> A2: It is crucial to highlight that **the main advantages of our proposed methods are that triCL 1) enhances the identifiability and interpretability of features (improving performance on feature importance relevant tasks), and 2) keeps comparable performance on feature importance irrelevant tasks,** compared with conventional contrastive learning.
>
> For **feature importance relevant tasks**, the experiments in Section 6.2 show that triCL achieves significantly superior k-NN accuracy, linear accuracy, and image retrieval precision on selected dimensions, which verifies that the importance matrix $S$ can automatically find the most important features.
>
> For **feature importance irrelevant tasks**, the experiments in Section 6.3 show that triCL can achieve comparable performance as the original contrastive learning. As Theorem 5.1 shows that triCL has an equal generalization bound to the 2-factor spectral contrastive loss [a], the comparable performance is expected and consistent with the theoretical analysis.
>
> Besides, following your suggestions, we conduct additional experiments to compare the training process of triCL and the original contrastive learning. To be specific, we compare the training loss of SimCLR and tri-SimCLR on ImageNet-100. **As shown in Figure A in rebuttal pdf, we observe that the two objectives have similar training curves and converge at a similar pace.** And there is no evidence that triCL makes the training easier stuck in local minima.
>
> [a] Jeff Z HaoChen, Colin Wei, Adrien Gaidon, and Tengyu Ma. Provable guarantees for self-supervised deep learning with spectral contrastive loss. In NeurIPS, 2021.
>
> ---
>
> Q3: Limitations.
>
> A3:  The primary advantages of our proposed tri-contrastive learning approach are the enhanced feature identifiability and interpretability with comparable performance in feature importance irrelevant tasks. This method proves to be particularly beneficial for tasks that require identifiable and explainable features, such as image retrieval tasks. However, when the downstream tasks are irrelevant to the interpretability of learned representations (e.g., linear probing and finetuning), triCL only achieves comparable performance as conventional contrastive learning. In the future, we will explore more tasks that need feature interpretability and expand the scope of the application of triCL.

---

> > ### Comment · Reviewer_tkty · 2023-08-18
> > **Thanks for authors’ rebuttal**
> >
> > Q1. This addresses my concern.
> >
> > Q2. I am not so convincing about the answers. I think the authors should show something that supports the proposed method is necessary for learning feature importance in training together with contrastive learning. Maybe there is another experiment that is needed: Feed the features from a learned contrastive learning model to a feature importance learning algorithm as a post-processing step, and compare the results.
> >
> > I agree with Reviewer aafx that i do not see the evidence in the experiments to support the theoretical claim on generalization, and the rebuttal seems not to provide more evidence neither.

---

> > > ### Author Response · Authors · 2023-08-19
> > > **Reply to responses**
> > >
> > > Dear Reviewer tkty, thanks for your responses. We will address your further concerns in the following.
> > >
> > > ---
> > >
> > > Q4. Another experiment is needed: Feed the features from a learned contrastive learning model to a feature importance learning algorithm as a post-processing step, and compare the results.
> > >
> > >  A4. Thank you for the insightful suggestion to employ post-processing feature importance estimation techniques on pretrained features. In response, **we have incorporated two feature importance learning algorithms for our evaluation: PCA and PRoFILE [a]**. For comparison with our proposed triCL method, we applied both PCA and PRoFILE to features generated by spectral contrastive learning (SCL), specifically selecting the 20 most important dimensions. Downstream linear evaluations were then conducted on these chosen dimensions.
> > >
> > > *Table A: The linear accuracy of selected dimensions of triCL, PCA, PRoFILE, and SCL on CIFAR-10.*
> > >
> > > |  | triCL (20 dimensions with largest S values) | PCA (20 dimensions with largest singular values in SVD decomposition) | PRoFILE  (20 dimensions with largest estimated importance) | SCL (20 random dimensions) |
> > > | --- | --- | --- | --- | --- |
> > > | Linear Accuracy  | 82.0 | 61.1 | 74.3 | 69.0 |
> > >
> > > As shown in the above table, we observe that **the features selected by the importance matrix show much better linear performance than the features selected by PCA and PRoFILE**.  We observe that PCA even performs worse than random selections (61.1% vs 69.0%), probably because the representations learned by SCL are nonlinear where PCA would fail to find the important dimensions. As for PRoFILE, although it shows better performance than random selections, triCL still achieves significant improvements over PRoFILE (82.0% vs 74.3%). This analysis leads us to believe that post-processing feature importance learning methods may not find the most important pretrained features accurately, thus highlighting the benefits of our triCL approach.
> > >
> > > [a] Accurate and Robust Feature Importance Estimation under Distribution Shifts, AAAI21.
> > >
> > > Q5. The experiments cannot support the theoretical claim on generalization.
> > >
> > > A5. It appears there might be some misconceptions regarding the primary focus of our paper. **Our main objective is not to advance generalization but to enhance feature interpretability and identifiability while maintaining comparable generalization performance.**
> > >
> > > To clarify, **our theoretical assertion in Theorem 5.2 states that triCL has the same theoretical guarantee as spectral contrastive learning for the linear probing task** (as mentioned in lines L205-L209 of the paper). This is further supported by the comparable downstream performance in feature importance irrelevant tasks between the two contrastive objectives, which aligns with our theoretical analysis.
> > >
> > > Moreover, **the primary strengths of our proposed methods lie in feature identifiability and interpretability.** This is exemplified in **Section 6.2**, where our experiments demonstrate that triCL significantly outperforms in terms of k-NN accuracy, linear accuracy, and image retrieval precision on selected dimensions.
> > >
> > > ---
> > >
> > > Hope our elaborations and new results above could address your concerns. Please let us know if there is more to clarify.

---

> > > > ### Comment · Reviewer_tkty · 2023-08-20
> > > >
> > > > Thanks for the reply. I do not have further questions, and I'd like raise my rating.

---

> ### Author Response · Authors · 2023-08-17
> **Your invaluable input is needed**
>
> Dear Reviewer tkty, thanks for your time reviewing our paper. We have meticulously prepared a detailed response addressing the concerns you raised. Could you please have a look to see if there are further questions? Your invaluable input is greatly appreciated. Thank you once again, and we hope you have a wonderful day!

---

### Official Review · Reviewer_eV59 · 2023-07-07

**Soundness:** 3 good
**Presentation:** 3 good
**Contribution:** 3 good
**Rating:** 7
**Confidence:** 4

**Summary:**

The paper proposes a new contrastive learning objective that learns an additional diagonal matrix which forces the features to exhibit different importance. Theoretical results are provided to show that the new algorithm achieves similar downstream linear probe accuracy but also better interpretability / identifiability. Empirically, the authors test the new objective on several standard benchmark datasets and show that learned features exhibit better k-nn accuracy when a subset of dimensions are used, and the downstream linear probe accuracy can also be improved in certain cases.

**Strengths:**

- The motivation is clear and the paper is written well.
- It's nice that the learned representations can have a clear subset of dimensions that captures most of the semantic information. This can be especially useful in many retrieval use cases where the dimensionality is the key bottleneck for scaling up the size of data. With this modified representation, people can easily select a subset of dimensions to construct less informative but more computational efficient representations for downstream use cases.
- The proposed method is justified with corresponding theory.

**Weaknesses:**

- Only small datasets are used. Could include larger ones like ImageNet.
- Theorem 5.1 seems to be exactly following that in [HaoChen et al], so I'm not sure there's enough novelty in this specific theorem.

**Questions:**

Is there a easy way to project the full feature from SCL or SimCLR to a 20-dimensional subspace and still match the same k-nn performance of the top 20 dimensions from tri-SCL? I wonder whether doing some PCA can achieve that.

---

> ### Author Rebuttal · Authors · 2023-08-09
>
> We thank Reviewer eV59 for appreciating our work on its theoretical and empirical insights. We will address the main weaknesses and questions you mentioned.
>
> Q1: More experiments on the large-scale dataset ImageNet.
>
> A1: We conduct additional experiments on the large-scale dataset ImageNet. To be specific, we pretrain ResNet-18 with SimCLR and tri-SimCLR for 100 epochs on ImageNet. Following the settings of the downstream tasks on ImageNet-100 in Section 5.2, we examine whether tri-SimCLR can obtain ordered representations on ImageNet.
>
> *Table A: The comparison between SimCLR and Tri-SimCLR in importance-relevant tasks on ImageNet.*
>
> |  | SimCLR (Random 40 dimensions) | Tri-SimCLR (40 dimensions with largest S values) |
> | --- | --- | --- |
> | Linear Accuracy | 28.9 | **34.1(+5.2)** |
> | Retrieval mAP | 15.4 | **22.7(+7.3)**|
>
> As shown in the above table, in both linear evaluations and image retrieval with selected 40 dimensions, tri-SimCLR accurately orders the feature importance and achieves significant improvements compared to the original contrastive learning methods. The empirical results on ImageNet further verify that tri-contrastive learning enhances the interpretability of the learned representations on large-scale datasets. And we will add the results to the paper in future updates.
>
> ---
>
> Q2: Theorem 5.1 seems to be exactly following that in [HaoChen et al], so I'm not sure there's enough novelty in this specific theorem.
>
> A2: We note that the main advantage of the proposed triCL is that we establish the feature identifiability and enhance interpretability that does not hold for canonical CL features. Meanwhile, for feature importance irrelevant downstream tasks, we prove in Theorem 5.1 that it has the same generalization guarantees as canonical CL following Haochen et al. It is worth mentioning that this theorem mainly serves as a sanity check that ensures its generalization ability, which is not the key contribution of this work. In more details, although following the same vene, our results generalize Haochen et al by showing that the conventional 2-factor contrastive loss is not the only choice, and a tri-contrastive loss can also attain an equal generalization bound. Thus, we believe that the theorem is a necessary part of the paper, but not the key contribution.
>
> ---
>
> Q3: Is there an easy way to project the full feature from SCL or SimCLR to a 20-dimensional subspace and still match the same downstream performance of the top 20 dimensions from tri-SCL (e.g., doing some PCA)?
>
> A3: Indeed, it is an interesting idea to reduce the dimensions of pretrained features with PCA. To compare it with triCL, we conduct additional experiments by applying PCA to reduce the dimensions from 256 to 20. Then we conduct linear evaluations on the selected dimensions.
>
> *Table B: The linear accuracy of selected dimensions of triCL, SCL+PCA, and SCL on CIFAR-10.*
>
> |  | triCL | SCL (PCA features) | SCL (random) |
> | --- | --- | --- | --- |
> | Linear Accuracy  | **82.0** | 61.1 | 69.0 |
>
>  As shown in the above table, we observe that the features selected by the importance matrix show much better linear performance (82.0 vs 61.1) than the features generated by PCA.  And we observe that PCA even performs worse than random selections, probably because the representations learned by SCL are nonlinear and PCA would fail to find the important dimensions. So we suspect that simple dimension reduction methods like PCA cannot find the most important pretrained features accurately, which further shows the advantages of triCL.
>
> ---
>
> Hope our elaborations and new results above could address your concerns. Please let us know if there is more to clarify.

---

### Author Rebuttal · Authors · 2023-08-09

The Rebuttal PDF can be seen in the attached file, which contains

- Figure A: comparison of the training loss between SimCLR and tri-SimCLR;
- Figure B: an additional explanation of Figure 1 in the paper;
- Figure C: a visualization example of how the augmentation strength influences the labeling error $\alpha$ in Theorem 5.1.

---

### Comment · Area_Chair_ZRt6 · 2023-08-21
**Further Discussion**

Dear Reviewers,

The open discussion phase of the paper is nearing its end, and the authors have provided more detailed elaboration, ablation studies, and explanations of the principles in the rebuttal phase, mainly in response to the proposed algorithms. In order to ensure the smooth running of the conference, we would like to receive the response of each reviewer to the authors' rebuttals as soon as possible. Therefore, we kindly ask you to submit your feedback as early as possible, if possible. Once again, we thank you for your time and look forward to your valuable comments.

Best wishes,

AC of Paper ID10613

---

### Decision · Program_Chairs · 2023-09-21

**Decision:**

Accept (poster)

**Comment:**

The paper proposes a new contrastive triCL method, which extends the existing pair-wise contrastive framework. The core motivation is the inability of CL methods to deliver identifiable representations. Theoretical results are provided to show that the new algorithm achieves similar downstream linear probe accuracy but also better interpretability/identifiability. Empirically, the authors test the new objective on several standard benchmark datasets and show that learned features exhibit better k-nn accuracy when a subset of dimensions is used, and the downstream linear probe accuracy can also be improved in certain cases.

This paper is well-written and makes good contributions. After author-reviewer discussions and reviewer-ac discussions, the paper still received mixed scores with three positives and one negative. The main concern of Reviewer aafx is the generalization capability of feature importance. Please remember to carefully explain this point (as you replied to the reviewer) in the camera-ready version.